# Functional protein mining with conformal guarantees

Ron S. Boger [1,2,3], Seyone Chithrananda [2,4], Anastasios N. Angelopoulos [4,5], Peter H. Yoon [2,6], Michael I. Jordan [5,7] & Jennifer A. Doudna [2,3,6,8,9,10,11] ✉

Molecular structure prediction and homology detection offer promising paths to discovering protein function and evolutionary relationships. However, current approaches lack statistical reliability assurances, limiting their practical utility for selecting proteins for further experimental and in-silico characterization. To address this challenge, we introduce a statistically principled approach to protein search leveraging principles from conformal prediction, offering a framework that ensures statistical guarantees with user-specified risk and provides calibrated probabilities (rather than raw ML scores) for any protein search model. Our method (1) lets users select many biologically-relevant loss metrics (i.e. false discovery rate) and assigns reliable functional probabilities for annotating genes of unknown function; (2) achieves state-of-the-art performance in enzyme classification without training new models; and (3) robustly and rapidly pre-filters proteins for computationally intensive structural alignment algorithms. Our framework enhances the reliability of protein homology detection and enables the discovery of uncharacterized proteins with likely desirable functional properties.

In the era of protein structure prediction, there are abundant opportunities for functional annotation of proteins. However, few robust methods available to introspect and assess the quality of these annotations, which is critical for selecting which proteins to characterize further through experimental or in-silico methods. Protein homology plays a central role in functional annotation, providing essential insights into protein functions and evolutionary trajectories. Protein homologs are proteins that share a common evolutionary origin, often displaying similarities in sequence, structure, or function due to gene duplication or speciation events. Homology provides a valuable framework for predicting the function of newly discovered proteins and understanding the molecular mechanisms underlying various biological processes. Homology searches generate a score indicating similarity between a query protein and proteins in a lookup database, based on either primary sequence or three-dimensional structural comparison.

Traditionally, homology search has focused on sequence comparison due to its speed and the limited number of experimentally solved protein structures. BLAST[1] and Hidden Markov Models (HMMs) have long been used to search large databases of protein sequences by scoring by residue overlap and alignment-based features. Classical methods for comparing protein structures such as DALI[2] and TM-align[3] confer higher sensitivity for finding remote homologs—protein homologs with low sequence similarity. However, these methods were not widely used due to the limited number of available protein structures and their slow speeds. With the development of accurate protein

[1]Biophysics Graduate Group, University of California, Berkeley, Berkeley, CA, USA. [2]Innovative Genomics Institute; University of California, Berkeley, CA, USA. [3]California Institute for Quantitative Biosciences, University of California, Berkeley, Berkeley, CA, USA. [4]Department of Electrical Engineering and Computer Sciences, University of California, Berkeley, Berkeley, CA, USA. [5]Department of Statistics, University of California, Berkeley, Berkeley, CA, USA. [6]Department of Molecular and Cell Biology, University of California, Berkeley, Berkeley, CA, USA. [7]Howard Hughes Medical Institute, University of California, Berkeley, Berkeley, CA, USA. [8]Molecular Biophysics and Integrated Bioimaging Division, Lawrence Berkeley National Laboratory, Berkeley, CA, USA. [9]Department of Chemistry, University of California, Berkeley, Berkeley, CA, USA. [10]Gladstone Institutes, San Francisco, CA, USA. [11]Gladstone-UCSF Institute of Genomic Immunology, San Francisco, CA, USA. ✉e-mail: doudna@berkeley.edu

structure prediction methods such as AlphaFold2[4], the number of available (predicted) protein structures has vastly increased. Despite this, large-scale searches through these predicted structures with classical structural alignment methods remain computationally infeasible.

Further, it is important to note that sequence and structural similarity do not necessarily imply shared function. For instance, there are enzymes for which functional annotation cannot be transferred to the protein of unknown function even when their pairwise sequence identity is greater than 90%, indicating different functions despite high sequence similarity[5,6]. Similarly, there are pairs of structures in the PDB with a TM-score greater than 0.5 and sequence identity below 10% that exhibit entirely different functions. Additionally, distant and meaningful homologies are often missed due to challenges such as long evolutionary distances, long-branch effects, and events like horizontal gene transfer that recombine protein parts and disrupt genomic context. Some estimates suggest that current methods may fail to detect more than half of all true homologous relationships between proteins, particularly at large evolutionary distances[7]. This underscores the need for methods that can more reliably infer function beyond sequence and structural comparison.

Approaches leveraging deep learning models on sequence, structure, and function such as TM-Vec[8], Foldseek[9], Protein-Vec[10], and TOPH[11] have provided a promising alternative for fast and highly sensitive homology search, outperforming classical methods on speed and nearly matching their sensitivity in traditional bioinformatic benchmarks. The practical application of these protein homology models, however, presents additional challenges. For example, a recent work Protein-Vec[10] presented state-of-the-art results across numerous benchmarks for function prediction. However, nearly all scores generated by this model fall within a range of 0.9995 to 1. These results are hard to interpret for a biologist because even normalizing the scores does not indicate which proteins are worthwhile for further characterization. Other approaches suffer from a similar problem of arbitrary thresholds; for instance, prior work has examined the significance of TM-align scores > 0.5 in relation to protein folds[12], but the selection of 0.5 is both arbitrary and does not provide statistical guarantees. This highlights the need for more non-arbitrary and reliable scoring systems that can guide experimentalists in selecting proteins for further investigation. For instance, a biologist may want to conduct a protein search that guarantees that 90% of the returned set shares biochemical function with the query protein (i.e., a 10% false discovery rate) and provides the probability of shared biochemical function within this set. The need for statistically valid "needle in a haystack" approaches to filter and retrieve a high-quality set of biological systems becomes increasingly important as genomic datasets rapidly expand. These challenges underscore the need for principled statistical methods like conformal prediction, which can provide statistically valid guarantees for protein retrieval and functional annotation.

Recent advances in conformal prediction offer a principled approach to protein retrieval, providing statistically valid and non-arbitrary prediction sets. Classical statistical techniques often rely on stringent assumptions about model structure (e.g., linearity) in order for their validity to hold; but in the era of deep-learning-based black-box models, such model assumptions rarely hold. Conformal prediction provides statistical guarantees that are black-box—they make no assumptions whatsoever about the structure of the model. These techniques address the emerging challenges that have arisen with the large scale of protein data and complex deep learning architectures, which cannot be addressed by statistical methods such as e-values and likelihood thresholds. So long as there is a calibration dataset representative of future data, conformal prediction offers a framework for returning sets of predictions with calibrated risk, such as false discovery rates or partial errors in an enzyme function annotation. By applying conformal prediction to homology search and function annotation models, we can transform raw similarity scores into retrieval sets and probabilities. Thus, we allow any search model to be employed for generating candidate homologs while providing a baseline guarantee of statistical accuracy for the final sets of proteins returned. Additionally, we convert raw similarity scores into calibrated probability estimates, providing normalized probabilities instead of raw scores. This enhances the interpretability of the model outputs, making them more accessible and useful for large-scale biological discovery.

In this work, we address the problem of assessing which proteins to characterize by developing a practical framework for reliable and interpretable evaluations, providing essential screening methods before costly and time-intensive biochemical or computational characterization. Specifically, we introduce a statistically principled framework leveraging conformal prediction for protein retrieval, focusing on efficiently returning high-quality sets of proteins with functional similarity with statistical guarantees on their validity. Our method adaptively ensures coverage and reliability, enhancing the interpretability of similarity scores while providing robust statistical guarantees for identifying proteins with homology and desired functional characteristics. Our approach helps determine which proteins should be further characterized in the lab or through higher-resolution algorithms, such as structural alignment or molecular dynamics. We demonstrate the utility of our approach across diverse protein datasets and tasks, from annotating genes of unknown function in the minimal viable genome to improving enzyme function prediction. Ultimately, our work reframes and addresses these challenges in quantitative biology as calibration and risk control problems, providing a robust approach for functional protein mining.

To further motivate the work, suppose the following setup. Assume we are given a set of query proteins $Q$ (i.e. uncharacterized proteins from a novel organism) and a large lookup database of proteins denoted $D$ (i.e. proteins which were previously experimentally characterized), and we use some statistical model of proteins to produce a set of similarity scores between the query and lookup protein, $S_{ij}$ for every $i \in [|Q|]$ and $j \in [|D|]$ (see Fig. 1A). For every query protein $q \in Q$, we aim to retrieve a subset of the lookup database $D$ that contains an attribute of interest to the query—for example, a protein from $D$ with sufficient functional similarity. With some abuse of notation, we sometimes refer to $Q_i$ and $D_j$ as the $i$th query protein from $Q$ and the $j$th lookup protein from $D$ respectively (with respect to some a-priori fixed ordering). This general problem setup is called retrieval.

Developing a selection method, i.e., a method for subsetting $D$, can be challenging because of errors in the model. In some cases, the model may assign lower similarity scores to proteins with functional similarity, while functionally distant proteins may receive higher scores due to model limitations. Thus, we aim to create a method that can find the optimal threshold $\hat{\lambda}$ based on a user-specified risk tolerance $\alpha$ for a desired loss metric for functional homologs, such as false negative rate (FNR) or false discovery rate (FDR) (Fig. 1). Given a query protein $q \in Q$ and a lookup set $D$, our method returns a set, $C_{\hat{\lambda}}(Q_i) = \{D_j \in D : S_{ij} > \hat{\lambda}\}$, where $\hat{\lambda}$ is selected using calibration data in a way that guarantees low risk.

The statistical techniques herein build on tools from the literature surrounding conformal prediction, as developed by[13,14]; for an introduction to contemporary techniques in the area, see[15]. Conformal prediction is a technique for calibrating arbitrary black-box prediction algorithms in order to satisfy statistical guarantees of marginal coverage (and notably, not conditional coverage[16,17]). This has become especially interesting in the era of deep learning, when prediction systems may be difficult to analyze with standard analytical statistics[18,19]. Here, we focus specifically on the use of conformal risk control techniques[20–22] for the purpose of biological retrieval algorithms. Although conformal prediction has been applied in several

ways to the biomedical space (see, e.g.,[23–31]), we are unaware of substantial work that resembles ours. A recent work published as we were writing this manuscript,[32], leverages conformal prediction to improve a machine learning model that the authors train, PenLight2. This is similar to section 3.2 in our work where we improve an existing machine learning model with better selection. Overall, however, our goal is not to present a machine-learning model for enzyme classification, but rather to develop a practical and rigorous methodology for experimentalists to decide what to characterize in the lab. Similarly, although some initial work has been done on conformal prediction and recommendation systems[33–36], none of these recommendation system techniques have been used with biological data to our knowledge.

## Results

### Annotation of genes of unknown function with control of the false discovery rate

Protein families (Pfams) are groups of evolutionarily related proteins that share a common ancestor. Members of a protein family typically have similar sequences, structures, and functions. Annotating protein families is critical in understanding their function and evolutionary history. Proteins can have multiple Pfam annotations; for instance, the bacterial immune system CRISPR-Cas9 is annotated with five Pfams for different functional domains (endonuclease, PAM interaction, etc.). The Pfam database is widely used in particular to classify protein

sequences into families in domains, and serves as a classic benchmark in functional annotation. We searched across Pfam-annotated proteins in Uniprot for exact (proteins where the Pfams are identical) functional matches using Protein-Vec and demonstrated our methods in finding the optimal similarity thresholds for false negative rate (FNR) and false discovery rate (FDR) at $\alpha = 0.1$. We shuffled the data over 100 trials to generate a new calibration dataset to learn optimal thresholds for FNR and FDR. It is also possible to learn optimal thresholds for partial (proteins sharing at least one Pfam) functional matches (see Section 5.2 for more details).

We assign probabilities of a functional match to each similarity score between query and lookup by fitting an isotonic regression. Isotonic regression is a nonparametric technique that fits a non-decreasing function to the data, allowing us to transform raw similarity scores into calibrated probabilities. This approach ensures that the assigned probabilities are monotonically increasing with respect to the similarity scores, a transformation that is a natural first step when assessing whether a given match is correct (Fig. 1). We employ an extended version of isotonic regression called Venn-Abers prediction[37], which comes with theoretical guarantees of calibration; see Section 5.1 for details, and Supplementary Fig. 1 for evaluations.

To evaluate the statistical validity of the isotonic regression, we employ Venn-Abers Predictors[37]. Venn-Abers Predictors are a type of conformal predictor that provides reliable prediction intervals,

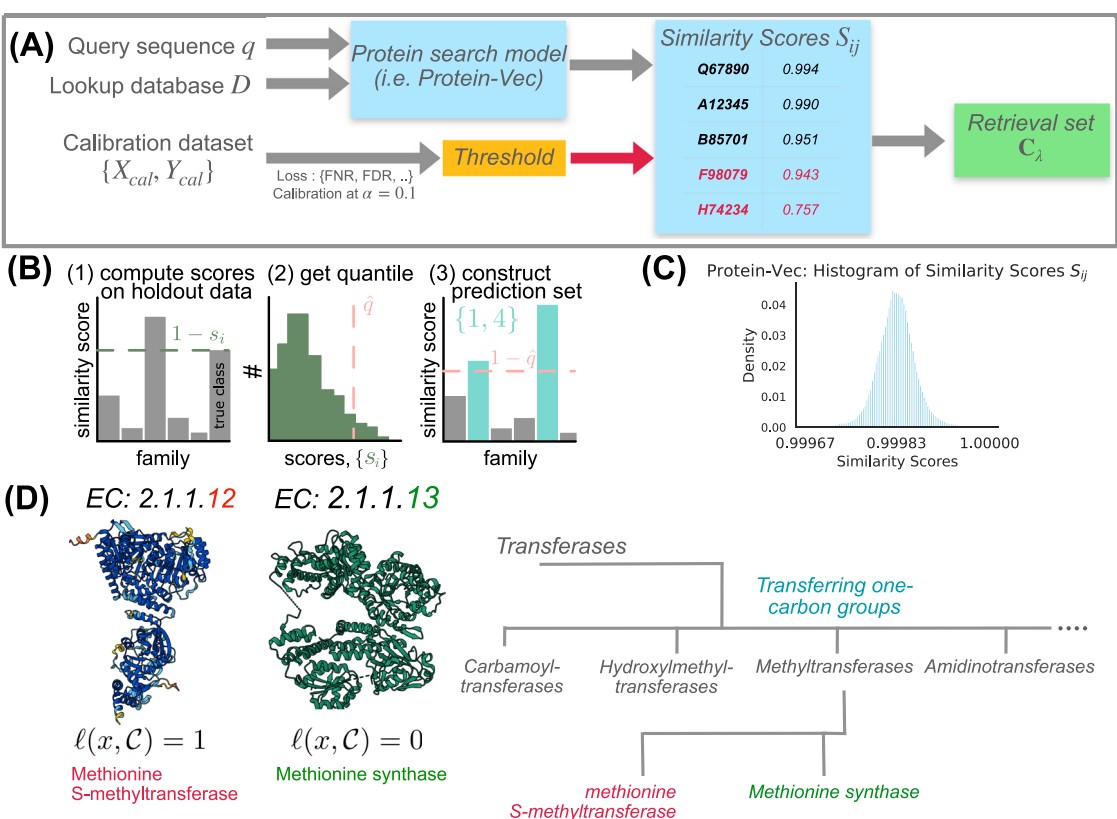

**Fig. 1 | Study Design and Motivation for Protein Homology Search Using Conformal Prediction. A** A query sequence $q$ is compared against a lookup database $D$ using a protein search model (e.g., Protein-Vec). The model generates similarity scores $S_{ij}$, which are compared against a threshold $\hat{\lambda}$ determined through calibration. Scores above the threshold are included in the retrieval set $C_{\hat{\lambda}}$. Scores below the threshold (e.g., F98079 with 0.943) are highlighted in red to indicate their exclusion. **B** The process involves computing scores on calibration data, obtaining quantiles, and constructing prediction sets. This approach provides statistical guarantees on the validity of the returned sets, enhancing the interpretability and reliability of protein search results. **C** The distribution of Protein-Vec similarity scores for UniProt motivates the need for effective thresholds and confidence

measures in protein homology searches, particularly given the high similarity scores clustering near 1. **D** Illustration of the error loss calculation for two enzymes: EC 2.1.1.12 (Methionine S-methyltransferase) and EC 2.1.1.13 (Methionine synthase). The loss function $\ell(q, C)$ assigns a value based on the maximum hierarchical loss of the enzymes in a retrieval set $C \subseteq D$, with 0 meaning every retrieved protein is an exact match. The hierarchical classification tree for part of transferases (EC 2) is shown, with methionine synthase being the ground-truth EC number, and methionine S-methyltransferase being in the model-retrieved retrieval set. This results in a $\ell(q, C) = 1$ hierarchical loss, due to a 4th-level family mismatch. Source data are provided as a Source Data file.

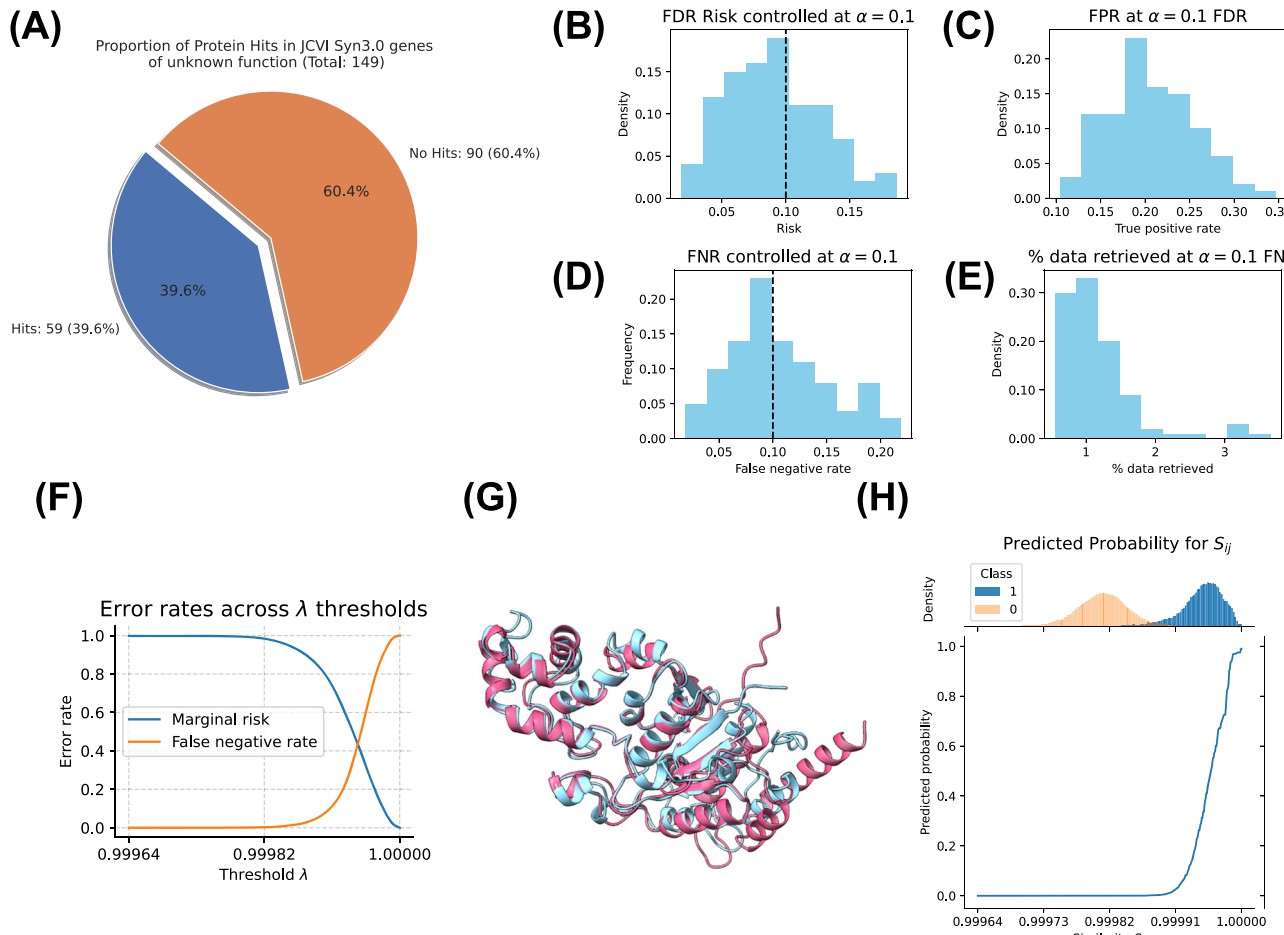

**Fig. 2 | Robust calibration of risk and probability for Pfam domain searches.**
**A** Hits represent proteins with similarity score above $\lambda$, determined by controlling the FDR at $\alpha = 0.1$. This yields exact functional hits for 39.6% of un-annotated genes in JCVI Syn3.0 *Mycoplasma mycoides*. **B** We control FDR on exact Pfam matches to $\alpha = 0.1$ and demonstrate calibration across 100 trials. **C** FDR control at $\alpha = 0.1$ retrieves roughly 25% of true positives. **D** We control for a false negative rate (FNR) loss at $\alpha = 0.1$ and demonstrate $\hat{\alpha}$ is well calibrated across 100 trials. **E** Using the threshold $\lambda$ controlling for FNR, we are able to reduce database size by 99% on

average. **F** Plot of false negative rates (FNR) and false discovery rates (FDR) as a function of similarity score threshold $\lambda$. As expected, FDR decreases as $\lambda \rightarrow \max(S_{ij})$ and FNR increases as $\lambda \rightarrow \max(S_{ij})$. **G** Structural alignment between predicted structure of functional hit of previously unannotated protein in *Mycoplasma mycoides* and characterized exonuclease. **H** Venn-Abers predictors assign probability of exact Pfam match (two proteins that share the same set of Pfams) given scores $S_{ij}$. Source data are provided as a Source Data file.

calibrated probabilities, adaptability to different loss functions, theoretical guarantees, and ease of implementation. This method helps verify that our probability assignments are statistically valid and that they maintain the desired coverage properties. By using Venn-Abers Predictors, we can ensure that the isotonic regression model produces accurate and reliable probability estimates for our similarity scores. We examine the difference in Venn-Abers test probabilities $\hat{p}^0, \hat{p}^1$ (the predicted probabilities of two isotonic regressions trained with different statistical parameters, see 5.1 for more details) and see that $|\hat{p}^0 - \hat{p}^1| \approx 0$, demonstrating that our model's ability to assess the probability of a functional match is well calibrated. We also show that the expected calibration error (ECE) of the Venn-Abers predictor is low, further indicating reliability (Supplementary Fig. 1). As a test case, we investigated the possibility of rigorously annotating genes identified in JCVI Syn3.0 *Mycoplasma mycoides*. JCVI Syn3.0, developed by the J. Craig Venter Institute, represents a minimal viable genome containing only the essential genes necessary for life[38]. Interestingly, despite its small genome, nearly 20% of the protein-coding genes in JCVI Syn3.0 were classified as genes of unknown function −genes with no homology to characterized genes via BLAST and HMMSearch − at time of publication. Annotating the genes in this synthetic organism is crucial for understanding their functions and the minimal

requirements for cellular life. Given the development in protein structure prediction and remote homology algorithms since the initial release of JCVI Syn3.0, we hypothesized that some fraction of these genes may have remote homology (that was not found via traditional methods) to well-characterized proteins. We applied our calibrated methods to this dataset, aiming to identify functional annotations for the previously unknown genes in Syn3.0. We assigned similarity scores to each of the genes of unknown function to Uniprot using Protein-Vec, and then filtered the results by selecting only $S_{ij} \geq \hat{\lambda}$, where $\hat{\lambda}$ is a threshold fit to obtain FDR control at $\alpha = 0.1$ (10% false discoveries expected). We find that 39.6% of coding genes of previously unknown function meet our criteria for an exact functional match (Fig. 2A) to proteins in UniProt. We demonstrate a structural alignment between a predicted structure of a gene of previously unknown function that met our criteria and a UniProt reviewed (ID Q9KAV6) exonuclease (Fig. 2G). By leveraging our approach, we provide robust and reliable annotations for previously uncharacterized yet essential genes, thereby contributing to the deeper understanding of minimal genomes and synthetic biology. Our approach can be broadly applied to rapidly assign high-confidence annotations to any genome of a new or understudied organism, illuminating the discovery of biological function in the natural and synthetic world.

## Robust selection strategies for enzyme function prediction

In addition to discovering genes of unknown function, we explore techniques for accurately annotating enzyme functions. Enzyme function annotation is a fundamental challenge in bioinformatics, critical for systems biology level understanding of metabolic pathways, drug development, and materials science. However, this task is inherently difficult because proteins can exhibit multiple enzymatic activities or none at all, and their functions can be influenced by complex structural and environmental factors. Traditional annotation methods often struggle with this complexity, leading to incomplete or inaccurate predictions.

To address these challenges, we explore a selection approach for a recent deep-learning model called CLEAN (contrastive learning-enabled enzyme annotation)[39]. CLEAN, akin to Protein-Vec, learns an embedding space for enzymes by employing a single-aspect contrastive loss function that minimizes the distance between similar (anchor and positive) enzymes, while maximizing the distance between dissimilar (anchor and negative) ones. CLEAN is based on Enzyme Commission (EC) numbers, a hierarchical numerical classification scheme for enzymes in which the catalytic function of an enzyme is specified by a series of four digits in increasing specificity. Using the learned embeddings from CLEAN, a two-component Gaussian mixture model is fit on the raw Euclidean distances between individual enzyme sequence embeddings and different EC number cluster embeddings. These EC cluster embeddings are computed using the mean embeddings across all sequences in the training dataset which have been annotated with the EC number, forming a centroid for the class. At inference time, two selection methods are used to predict EC numbers for enzyme sequences.

1. max $-$sep (*max-separation*), is a greedy approach that selects EC numbers with the maximum separation that stands out from other centroid embeddings.
2. p $-$ value (*p-value selection*), identifies EC number centroid similarity scores that stand out against the background distribution of $n = 20,000$ randomly sampled training similarity scores.

CLEAN has been evaluated on two independent datasets not included in the model's development to deliver a fair and rigorous benchmark study. The first, *New-392*[39], uses a date-cutoff on Uniprot to select 392 enzyme sequences covering 177 different EC numbers, containing data from Swiss-Prot released after CLEAN was trained (April 2022). The second, *Price-149*, was a set of experimentally validated results described by Price et al[40]. Curated by ProteInfer[41], *Price-149* is considered a challenging dataset because the existing sequences have frequently been incorrectly or inconsistently labeled in databases by automated annotation methods. Adding to this challenge is major data imbalances in the training data within UniProt, where we observe a strong left skew in the histogram of EC labels towards a handful of EC families with high label abundance. We observe that 4498 of the 5242 total EC annotation's in CLEAN's training data have less than 50 protein examples, illustrated by Supplementary Fig. 4.

Despite the advancements CLEAN offers, selecting the correct enzymatic functions with statistical confidence remains non-trivial. Given the frequent misannotation in the field and the effort devoted to selection in CLEAN, we were interested in adapting our conformal procedures independently for each dataset to develop a statistically grounded selection technique. The hierarchical nature of the EC system, in which each enzyme sequence can be thought of as a leaf node in a tree, accorded well with our use of a hierarchical risk function (7). Additionally, we wanted to explore whether calibrating on one dataset and evaluating on the second would maintain the coverage guarantees. This approach could potentially produce a more performant selection method than the two strategies proposed by CLEAN.

A conformal-derived similarity threshold $\hat{\lambda}$, in contrast to query-specific annotations, would i) provide performance guarantees for divergence against the hierarchical classification for some $\alpha$, and ii) enable the model to output an empty set when it is uncertain about whether the protein should be classified with an EC number at all, an issue both max $-$sep and p $-$ value selection do not reconcile. We demonstrate an example of this selection pathology in Fig. 3. When asking CLEAN to annotate an antigen-binding fragment of a recently-developed SARS-COV-2 antibody (a protein that is evidently not an enzyme), both max $-$sep and p $-$ value return annotated sets (as max $-$sep must return at least one annotation). In contrast, conformal risk control appropriately returns an empty one. CLEAN employs a Gaussian Mixture Model to assign a level of confidence to the results. While this method can measure probability and manually determine a confidence threshold for EC annotation, it comes with limitations. For high-throughput applications like metagenomic enzyme mining, conformal guarantees implicitly resolve these model pathologies, providing a more robust solution. Consequently, we calibrated on $N = 380$ of the 392 query points provided by *New*, and report test precision score, recall score, F1-score, and area under curve (AUC) in addition to the hierarchical loss coverage on *Price-149*, following the metrics reported by CLEAN. Thus, we assembled distograms for both *New-392* and *Price-149* against all 5242 EC cluster embeddings in CLEAN's training set. We then computed per-query hierarchical loss scores to calibrate with conformal risk control.

Our findings indicate that the conformal selection strategy, using the same underlying embeddings produced by CLEAN, outperforms both max $-$sep and p $-$ value selection. Most excitingly, we find that not only does calibrating on *New-392* and evaluating on a subset of *New-392* outperform the CLEAN selection methods, but calibrating on *New-392* and testing on the more difficult *Price-149* benchmark containing previously hard-to-annotate enzymes of unknown function also yielded strong performance. We report results for both tasks in Tables 1–2.

Despite the datasets not being exchangeable and noting significant shifts in (i) the distribution of similarity scores produced by CLEAN and (ii) sequence identities to functional matches in the training set for both datasets (see Supplementary Figs. 5–6), our hierarchical risk calibration strategy still outperformed both prior selection methods.

We believe this early work raises the opportunity to use our method, conformal protein retrieval, on a withheld subset of training data as a central, large calibration dataset. This approach can then be extended to multiple, separate annotation tasks for robust and reliable selection, ultimately enhancing the accuracy of enzyme function prediction across diverse datasets.

## DALI prefiltering of diverse folds across the proteome

Further, we demonstrate how to do robust and fast screening prior to using high-resolution yet slow in-silico algorithms such as molecular dynamics or structural alignment. While embedding-based search methods have brought about the ability to do large-scale searches with improved sensitivity, structural alignment methods remain important due to their ability to provide detailed biochemical and functional insights. We explored the possibility of extending calibration on a task of related function, remote homology, and extend it to a broader task that inherently relies on the same shared structural knowledge. In this section, we aim to do so by building a robust prefiltering technique for running DALI structural alignments,

Classical methods for comparing protein structures like DALI[2] and TM-align[3] confer high sensitivity for finding remote homologs and output a structural alignment. Structural alignment is crucial for biochemists and molecular biologists because it often provides insights into functional relationships – such as the identification of active sites, binding interactions, and conformational changes – which are often

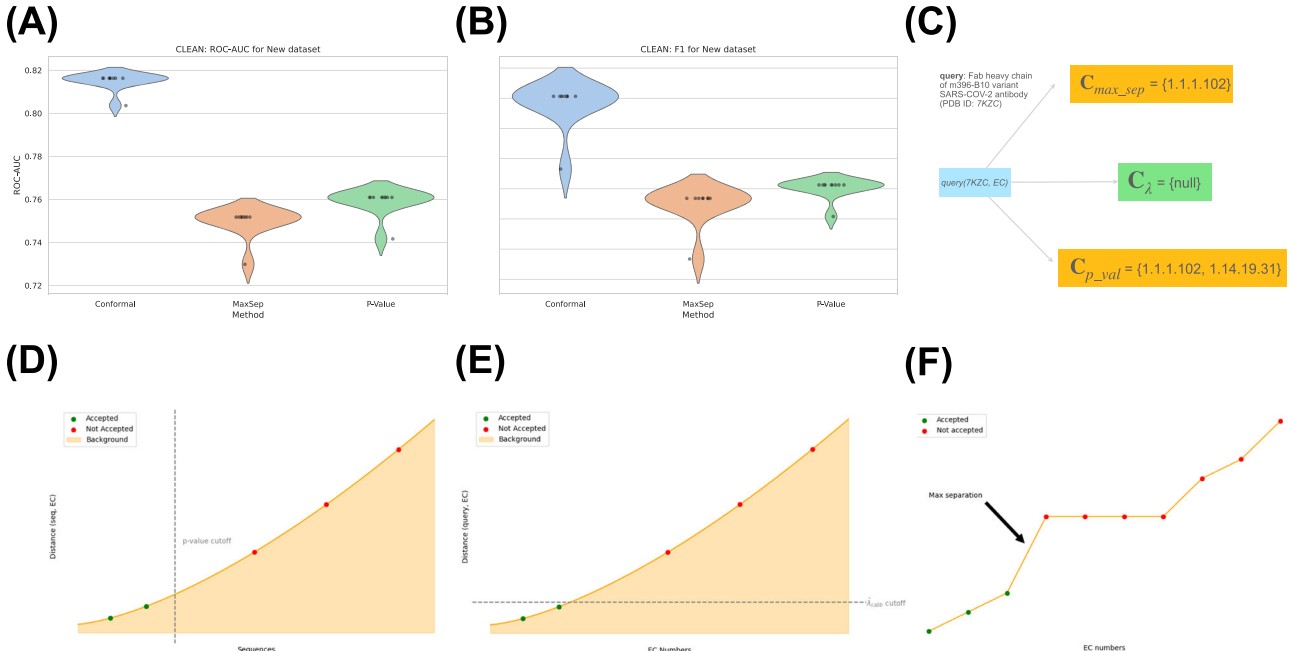

**Fig. 3 | Results for utilizing conformal prediction for enzyme function annotation, using a leading classification model.** We compare the two methods for "EC-calling" proposed by CLEAN[39], max−sep (max-separation) and p−value p-value selection, against our conformal method. We report confidence intervals through violin plots for 10 random shuffles of the dataset to ensure coverage across *New*. **A** Violin plots of ROC-AUC for conformal, p−value, and max−sep. **B** Violin plots of F1 for conformal, p−value, and max−sep. **C** Returned EC annotations for a SARS-CoV-2 antibody. Conformal selection correctly identifies SARS-CoV-2 antigen-binding fragments (Fabs) as not enzymes, whereas max−sep and p−value

methods from CLEAN return possible enzyme annotations. **D**−**F** Intuitive overview of selection methods for EC annotation using similarity scores. Here, **D** represents the cutoff threshold determined by p-values, which ranks query enzymes to each EC cluster center $EC_i$ amongst a background of random proteins from the training dataset. **E** displays our conformal distance-threshold, while **F** displays max-separation selection, which aims to select EC numbers that stand out from the other EC query-centroid distances. Here, the arrow describes the intuitive "point of max separation" amongst all EC numbers and the query. Source data are provided as a Source Data file.

**Table 1 | Comparison of CLEAN selection methods to conformal when calibrating on 300 data points and testing on 92 proteins from *New* (68–77 unique EC labels)**

| Method | Precision | Recall | F1 | AUC |
|---|---|---|---|---|
| **Conformal** ($\alpha = 1$) | **56.80 ± 1.64** | **63.71 ± 0.29** | **57.65 ± 1.45** | **81.50 ± 0.38** |
| Max-sep | 53.05 ± 0.13 | 50.03 ± 1.34 | 50.96 ± 1.21 | 74.96 ± 0.66 |
| P-Value | 53.27 ± 0.51 | 52.01 ± 1.11 | 52.04 ± 0.63 | 75.90 ± 0.58 |

Train and test indices are shuffled across 10 trials to ensure robust performance across different dataset partitions. Bold values indicate the best-performing method for each metric (higher is better).

**Table 2 | Generalizability test of CLEAN selection methods when calibrating on 380 data points from *New* and testing on all 149 samples from *Price* (56 EC classes)**

| Method | Precision | Recall | F1 | AUC |
|---|---|---|---|---|
| **Conformal** ($\alpha = 1.5$) | 55.98 | **49.34** | **49.62** | **74.59** |
| Max-sep | 58.44 | 46.71 | 49.47 | 73.34 |
| P-Value | **59.26** | 47.74 | 47.77 | 72.34 |

Nearly all of *New* is used for calibration, as increasing the size of the calibration set progressively tightens the loss coverage, detailed in Eq. (5). Bold values indicate the best-performing method for each metric (higher is better).

not evident from sequence comparisons alone. Traditionally, these structural alignment algorithms have been limited by the number of available protein structures and slower computational runtimes. Although accurate protein structure prediction methods such as AlphaFold2[4] have vastly increased the number of available (predicted) protein structures and structural databases broadly, these algorithms still remain computationally infeasible at scale. Computing a classical

structural alignment between a set of query proteins and large databases such as the AlphaFold Database (AFDB)[42], or even the recently clustered 2.3M subset of AFDB[43], requires time and specialized compute. For instance, a recent study discovered an ancestral CRISPR-Cas13 nuclease[44] using DALI structural alignment algorithm to search the clustered AFDB and released specialized software to aide in the search. For instance, computing an alignment for 73 SCOPe (Structural Classification of Proteins − extended)[45] protein domains against all of the clustered AFDB took ~ 1 day on our highly optimized super-computer setup with 10 threads (Table 3). As access to high-performance computational resources becomes more limited and the number of protein folds continues to expand, this problem will become more intractable. In contrast, embedding-based search methods such as Protein-Vec can perform the same task in only about ~ 30 seconds on a modern laptop. Therefore, it is valuable to select proteins efficiently prior to conducting computationally expensive in-silico analyses.

Motivated by these slow runtimes to get high-quality structural alignments and similarity scores, we choose to calibrate risk with a faster embedding-search model, Protein-Vec, on a related task of homology. We then evaluate its ability to retrieve high-ranking *Z*-score

hits from a DALI search as the ground-truth metric. Although tools have been developed to quickly estimate structural alignment scores[8], computationally expensive downstream analyses such as alignment and molecular dynamics will always be relevant for biologists. As such, developing a robust methodology to prefilter large databases of proteins into smaller subsets that retain a majority of proteins with desired biochemical functions remains a critical challenge in the field.

Different datasets of proteins may contain diverse genes that have varying distributions of homologous sequences, depending on their protein family, superfamily, or fold. For instance in SCOPe 2.08[45] clustered at 40% sequence identity, some families contain > 100 proteins, while others contain two or only one categorized protein. When estimating a threshold $\hat{\lambda}$, it is important to ensure we account for adaptive set sizes in our retrieval set to filter large sequence and structure databases. In addition, some proteins may have no known homologs, so a model should return an empty search set. A threshold determined with conformal risk control allows for this while providing calibrated statistical guarantees.

To develop a prefilter for DALI structural alignments using a fast but perhaps less sensitive model as a surrogate with statistical guarantees enables us to quickly search large databases. With the surrogate model's predictions, we aim to infer some threshold $\hat{\lambda}$ that can obtain our subset $\hat{C} \subset D$ that contains hits of structural similarity to the query. With this subset $\hat{C}$, we can then perform structural alignments with DALI, thereby reducing the overall search and computation time to feasible levels. Specifically, we address the challenge of reducing the 2.3M proteins in clustered AFDB while maintaining a minimal FNR calibrated through conformal risk control. In effect, we seek to select the smallest possible set which captures nearly all high Z-score homologs that would have been identified by DALI in a comprehensive clustered AFDB search.

To measure our ability to do this, we demonstrate the use of conformal risk control to prefilter DALI across diverse "multi-domain" folds from the Structural Classification of Proteins—extended database (SCOPe)[45] (see 5.2 for selection procedure used in SCOPe prefilter strategy). We begin by embedding all proteins in SCOPe and the clustered AFDB using Protein-Vec, and use conformal risk control to learn a $\hat{\lambda}$ that achieves a 1% false negative rate (FNR) for SCOPe families. This calibration task is disjoint from the subsequent task, where we use the calibrated threshold to search and select proteins from the clustered AFDB. By doing so, we aim to retain nearly all high Z-score homologs identified by DALI. Although we use Protein-Vec for ease, we note this can be done with Foldseek and other fast models that mimic structural alignment scores.

We infer a threshold that may tolerate a higher FDR for low DALI Z-scores but ensures a low FNR for DALI Z-scores greater than $Z'$. We define the threshold $Z'$ as the elbow-point of the descending sorted Z scores per the Kneedle algorithm[46], representing the point where the rate of decrease in Z-scores starts to slow significantly. We consider partial matches as those diverging at the family but preserving superfamily-level homology. We then use the learned threshold, $\hat{\lambda}$, as a prefilter for the DALI multi-domain search task, obtaining a subset $\hat{C} \subset D$, where $D$ 2.3M protein clustered AFDB (see Section 5.2 for pre-processing steps).

We illustrate the effectiveness of our prefiltering approach in Fig. 4. We display the correlation between DALI and Protein-Vec similarity scores for SCOPe domains against the clustered AFDB. When looking at the correlation between DALI and Protein-Vec similarity scores for SCOPe domains against the clustered AFDB, two distinct takeaways emerge. First, retrieved hits with higher structural similarity scores $S_{ij}$ in Protein-Vec, corresponding to $Z - score \geq 10$ in DALI, exhibit a distinct distribution shift, indicating our method's ability to capture significant hits. In contrast, $Z - score \geq Z'$ exhibits much greater variance in Protein-Vec similarity scores. Despite this, we show that the retrieved $\hat{C}$, captures 82.8% of hits above $Z'$ from $D$, while filtering out 31.5% of the clustered AFDB (Tables 4–6).

These results demonstrate that our prefiltering method effectively reduces the size of the lookup database while retaining a majority of homologous proteins with desired biochemical properties. By significantly reducing the clustered AFDB set while maintaining a low FNR, our approach enables more efficient and feasible structural alignments with DALI. This strategy not only meaningfully reduces computational demands but also ensures the comprehensive identification of likely structural homologs, providing a valuable tool for structural biologists and biochemists to accelerate discovery.

## Discussion

The rapid increase in genomic data and the development of new algorithms for protein search mark an exciting time for computational biology. In this study, we demonstrate a robust approach for protein search that provides statistical guarantees on the retrieval of homologous proteins, thereby enabling principled prioritization for further biochemical and biophysical characterization. Our method is grounded in conformal prediction, which enables the transformation of raw similarity scores into interpretable retrieval sets and probabilities with

**Table 3 | Comparison of runtime between DALI and Protein-Vec for searching 73 SCOPe domains against the full clustered AFDB**

| Method | Runtime | Number of threads |
|---|---|---|
| DALI | ~ 1 day | 10 |
| Protein-Vec | ~ 30 seconds | 1 |

Given the substantial speed difference between Protein-Vec and prior methods, prefiltering reference databases before structural alignment can significantly reduce computational costs.

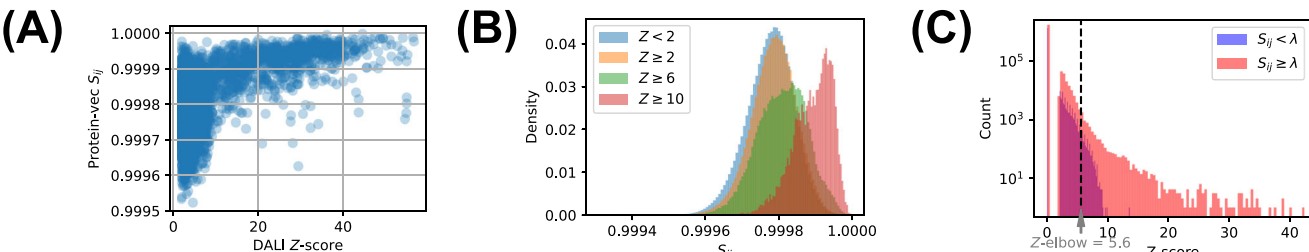

**Fig. 4 | Statistically robust prefiltering can reduce the size of a lookup database for high accuracy yet computationally intensive methods like DALI.**
**A** Correlation between DALI and Protein-Vec similarity scores for SCOPe domains against the clustered AFDB. Proteins with a DALI $Z < 2$ are reported as $Z = 0$, as they are not outputted by DALI. **B** Distribution of Protein-Vec scores with associated Z-scores at different Z-score values. There is an observable distribution shift as $Z$ increases. **C** Histogram of $Z$ scores below and above the learned threshold $\lambda$ to ensure $\alpha = 0.01$ FNR for SCOPe families. We observe that most of the distribution density for Protein-Vec scores $S_{ij} < \lambda$ is contained below $Z'$, the elbow in the Z-score distribution. Source data are provided as a Source Data file.

**Table 4 | Statistical summary of DALI search hits excluded when filtering Protein-Vec hits for thresholds $Z \geq Z\prime$, the Z-score elbow point found with the Kneedle algorithm**

|        | FNR   | TPR   | FDR   |
|--------|-------|-------|-------|
| Mean   | 0.182 | 0.818 | 0.997 |
| Std    | 0.174 | 0.174 | 0.004 |
| Min    | 0.000 | 0.290 | 0.979 |
| Median | 0.121 | 0.879 | 0.998 |
| Max    | 0.710 | 1.000 | 1.000 |

We report false negative rates (FNR), true positive rates (TPR), and false discovery rates (FDR).

**Table 5 | Statistical summary of DALI search hits excluded when filtering Protein-Vec hits for thresholds $Z \geq 2$**

|        | FNR   | TPR   | FDR   |
|--------|-------|-------|-------|
| Mean   | 0.282 | 0.718 | 0.943 |
| Std    | 0.236 | 0.236 | 0.070 |
| Min    | 0.010 | 0.108 | 0.712 |
| Median | 0.206 | 0.794 | 0.973 |
| Max    | 0.893 | 0.990 | 1.000 |

We report false negative rates (FNR), true positive rates (TPR), and false discovery rates (FDR).

**Table 6 | Statistical summary of DALI search hits excluded when filtering Protein-Vec hits**

|        | $\frac{N(S_{ij}>\lambda)}{N}$ | $\frac{N(Z<2)}{N}$ | $Z\prime$ |
|--------|-------|-------|-------|
| Mean   | 0.685 | 0.947 | 5.130 |
| Std    | 0.246 | 0.067 | 1.670 |
| Min    | 0.104 | 0.737 | 3.200 |
| Median | 0.768 | 0.974 | 5.000 |
| Max    | 0.978 | 1.000 | 14.600 |

Metrics include the percentage of the data retained at the $\hat{\lambda}$ threshold ($\frac{N(S_{ij}>\lambda)}{N}$), the percentage of hits with DALI structural alignment scores below 2 ($\frac{N(Z<2)}{N}$), and the $Z\prime$ elbow determined by the Kneedle algorithm.

statistical guarantees. Our approach for statistical guarantees in protein retrieval has meaningful applications across multiple areas in biochemistry, bioinformatics, and structural biology. Namely, we show statistically valid annotation of genes previously deemed as genes of unknown function, state-of-the-art performance on an enzyme classification task without training a deep learning model, and robust prefiltering of the clustered AlphaFold Database to reduce the computational burden for structural alignment.

Although we extensively use Protein-Vec in this work, our approach is model agnostic and can be used with any search or function annotation algorithm. Indeed, protein search methods that rely on embedding pairs of proteins and computing their cosine similarity via FAISS can especially benefit with additional statistical guarantees. It is conceivable that fast structural search algorithms such as Foldseek could show an even greater improvement in prefiltering the clustered AFDB for an individual query prior to performing a more intensive structural alignment. Additionally, we could better calibrate a prefilter threshold using DALI scores for our entire SCOPEe40 2.08 test vs lookup set, and calibrating to find some binned $Z -$ score threshold we wish to consistently retrieve in our confidence set. Such an approach would enable the calibration set to be truly *exchangeable* to the test DALI scores we are filtering against, and likely result in a higher filter and lower FNR rate.

Our approach is not without limitations. Conformal prediction assumes exchangeability, which may not apply to all protein homology searches. However, this is not always the case with proteins due to several factors. Organisms are sampled at different frequencies, leading to a bias where certain protein families are overrepresented, particularly those from more frequently studied organisms. When presented with new data, this may result in distribution shifts that are challenging for the underlying conformal approach. Additionally, the quality of samples can vary significantly, with metagenomic proteins often being of lower quality compared to those from well-characterized organisms. Furthermore, some protein families may be missing altogether, as humans have only sequenced a fraction of the life on Earth. Finally, these datasets often lack large individual variation; single amino acid changes can meaningfully alter a protein's active site and function, yet alignment-based methods may not fully capture these subtle but important differences. Calibration is only as good as the quantity and quality of labeled data, which underscores the importance of comprehensive and accurate datasets for achieving reliable results. There is no silver bullet to address such distribution shifts; however, there have been many recent advances in conformal prediction under distribution shift that can be brought to bear on the topic[47–50]. Extending our method using these techniques would be an interesting topic for future research, although it would increase the complexity of the calibration process.

We do not produce uncertainty bounds in the underlying similarity score for a pair of proteins, but instead uncertainty between a query and lookup database. For a model such as TM-Vec which estimates the structural TM-align scores for a pair of two protein sequences, methods such as conformal quantile regression[51] could be used to provide confidence bounds around the true TM-align score. Furthermore, given that many new protein search models leverage advances in protein language models, it may be advantageous to use the sequence perplexity derived from these language models. Indeed, it has been shown that protein language models are biased by unequal sequence sampling across the tree of life[52], which can result in higher sequence perplexity for other organisms. In addition, these methods offer marginal, not conditional, statistical guarantees, the learned thresholds might not generalize consistently to all protein classes.

In summary, our work represents a significant step forward in the field to move from protein search to experimental characterization by integrating statistical guarantees through conformal prediction. The ability to annotate genes of unknown function, classify enzymes with high accuracy, and reduce computational overhead for structural alignments highlights the practical benefits of our approach. As the volume and diversity of genomic data continue to expand, the need for reliable and efficient protein search methods becomes increasingly critical. Future advancements in conformal prediction and protein language models will further enhance the robustness and applicability of these methods, driving new discoveries and innovations in biology. Our framework addresses the challenge of screening and selection proteins for deeper characterization, paving the way for more reliable and efficient discovery of proteins with valuable functional properties.

## Methods
### Protein retrieval and conformal prediction formalisms
In this section, we overview the mathematical tools we use in order to perform retrieval for protein homologs. In particular, the key is various methods for constructing subsets of protein space which have guarantees of retrieval, but are small enough to narrow down the search process.

Throughout the appendix, we will switch to a more mathematical notation to parallel the statistics literature on these topics. We will let $\mathcal{X}$ and $\mathcal{Y}$ be the query and response (lookup) proteins, respectively. In the retrieval problem, we seek to associate each query $x \in \mathcal{X}$ with a set of responses. Given a set $\mathcal{C} \subseteq \mathcal{Y}$, we measure the quality of the set $\mathcal{C}$

using a loss function:

$$\ell(x, \mathcal{C}) \in [0, B] \text{ for some } B < \infty. \tag{1}$$

As an example of a loss function, imagine that for every the query protein $x \in \mathcal{X}$ and response protein $y \in \mathcal{Y}$, we can associate a degree of SCOPe ID matching, $\text{match}(x, y) \in \{0, 1, 2, 3, 4\}$. When $\text{match}(x, y) = 0$, it means the proteins are an *exact match*—that they each have a site with exactly the same SCOPe family. Meanwhile, when $\text{match}(x, y) = 4$, it means that the proteins are not even in the same SCOPe class. Intermediate levels of matching indicate intermediate SCOPe ID matches. With this match function in hand, we can define the loss function as, for example, the fraction of our retrieved set that is *not* exact matches. Mathematically, this loss function is

$$\ell(x, \mathcal{C}) = \frac{1}{|\mathcal{C}|} \sum_{y \in \mathcal{C}} \mathbb{1}\{\text{match}(x, y) \neq 0\}, \tag{2}$$

where $\mathbb{1}$ is the indicator function that takes the value 1 when the argument is true and 0 otherwise. We are trying to generate retrieval sets that ensure this loss is small.

Our main tool will be to index some family of sets by a one-dimensional parameter, $\lambda \in \mathbb{R}$. We will refer to this family of sets as $\mathcal{C}_\lambda$. An example would be

$$\mathcal{C}_\lambda(x) = \{y \in \mathcal{Y} : f(x, y) \geq \lambda\}, \tag{3}$$

where $f$ is a pre-trained machine learning model trained to predict whether $x$ and $y$ match. The methodologies exposed herein allow us to pick the parameter $\lambda$ such that these sets have a small loss in a probabilistic sense,

$$\mathbb{E}[\ell(X, \mathcal{C}_{\hat{\lambda}}(X))] \leq \alpha. \tag{4}$$

The parameter $\hat{\lambda}$ is picked based on a calibration procedure which uses a small dataset of proteins the model has not seen during training. We will call this calibration dataset $X_1, \ldots, X_n$, and we assume we can evaluate $\ell$ against any of the possible responses in $\mathcal{Y}$. Then, we will deploy the $\hat{\lambda}$ that is picked using this calibration data on a new, exchangeable protein $X_{\text{test}}$.

The critical assumption in all the forthcoming techniques is the *exchangeability* of the calibration data and the test point. Exchangeability means that the joint distribution function of the calibration data and the test data point is invariant to permutations. As an example, i.i.d. data points are exchangeable; exchangeability is a weaker condition than this. Intuitively, this means that the calibration data must be representative of the test data, and not involve any deleterious distribution shifts. The particular technical condition required for our theory is that the vector of losses $(\ell(X_1, \mathcal{C}_\lambda(X_1)), \ldots, \ell(X_n, \mathcal{C}_\lambda(X_n)), \ell(X, \mathcal{C}_\lambda(X)))$ is exchangeable for all $\lambda$—the exchangeability of the data points implies this fact, but it is technically weaker. In other words, exchangeability matters only insofar as the risk is concerned.

For clarity, we define some of the commonly utilized loss functions for our retrieved sets, false discovery rate (FDR) and false negative rate (FNR). Motivated by the desire to control against false significant hits, we define false discovery rate first. FDR measures the ratio between false positive hits *(false discoveries)* in our retrieved set of model-derived significant hits to the total number of hits (the size of our retrieved set). This is expressed as $\frac{FP}{FP+TP}$. The FNR, similarly, is the number of false negative significant hits (annotated hits not in the retrieval set) as a fraction of the total pool of possible hits, expressed as $\frac{FN}{FN+TP}$. For further literature relating to controlling FDR, we refer readers to[53].

**Conformal risk control.** Conformal risk control[21] is an extension of conformal prediction that provides an algorithm for satisfying (4) when the function $\lambda \mapsto \ell(x, \mathcal{C}_\lambda(x))$ is monotone for all $x$. As such, conformal risk control extends conformal prediction to control the expected value of any monotone loss function. The monotonicity is critical for the theoretical guarantee to hold, as $\lambda$ increasing should ensure the prediction sets become more conservative and that $\ell(x, \mathcal{C}_\lambda(x))$ does not incrase. The threshold function $\hat{\lambda}$ in the following way:

$$\hat{\lambda} = \inf\left\{\lambda : \frac{1}{n}\sum_{i=1}^{n} \ell(x, \mathcal{C}_\lambda(x)) \leq \alpha - \frac{1-\alpha}{n}\right\}. \tag{5}$$

When the data points $X_1, \ldots, X_n$ and $X$ are exchangeable, this results in exactly the guarantee in (4).

The calibration procedure for $\hat{\lambda}$ is doing something simple and easy to understand. On the left-hand side of the inequality in (4), we have the empirical risk, i.e., the average loss on our calibration data. On the right-hand side of the inequality, we have $\alpha$ minus a small fudge-factor that decays as $1/n$. Thus, we are picking the smallest $\lambda$—often indicating the smallest retrieval set—such that the risk is bounded above by $\alpha$ (fudge-factor aside). Importantly, the loss in (4) is not any specific loss, like the false negative rate or false discovery rate—it is a general, bounded loss (although non-monotone losses have a slightly different algorithm; see[22]).

**Hierarchical risk control.** Here, we explain how conformal risk control can be used to do hierarchical prediction of the SCOPe/CATH ID/EC of a protein. In other words, we will directly address the task of predicting a family for protein $X$. Protein families are normally classified in a hierarchy through SCOPe and CATH, with their place in the hierarchy represented as a vector:

$$h(x) = (A, B, C, D), \tag{6}$$

where $A$, $B$, $C$, and $D$ are strings corresponding to the domain, super-family, fold, and family, respectively. We let $\mathcal{C}$ take values in the space $\{A\} \cup \{A. B\} \cup \{A. B. C\} \cup \{A. B. C. D\}$, for all integers $A$, $B$, $C$, and $D$, respectively. Let our loss function be as follows:

$$\ell(x, \mathcal{C}) = c_{\min\{i \in [4] : h(x)_i = \mathcal{C}_i\}}, \tag{7}$$

where $c_i$ are nonnegative constants. An example would be $c_1 = 3$, $c_2 = 2$, $c_3 = 1$, and $c_4 = 0$, in which case the loss function reduces to $\text{match}(x, \mathcal{C})$. The key observation is that running conformal risk control at level $\alpha$ with the above loss results in the following property:

$$\mathbb{P}(\ell(X, \mathcal{C}_{\hat{\lambda}}(X)) = i) \leq \frac{\alpha}{c_i}, \tag{8}$$

for all $i$. (Here, $\alpha/0 = \infty$.) Intuitively, this means that conformal risk control can be used to simultaneously bound the probability of all mismatches, with a penalty that grows as the mismatches become more extreme. The proof of this property follows from the definition of the expected value:

$$\mathbb{E}[\ell(X, \mathcal{C}(X))] \leq \alpha \iff \sum_{i=1} c_i \mathbb{P}(\ell(X, \mathcal{C}(X)) = 4 - i) \leq \alpha$$
$$\Rightarrow c_i \mathbb{P}(\ell(X, \mathcal{C}(X)) = 4 - i) \leq \alpha, \forall i. \tag{9}$$

**Non-monotone risks.** In this paper, we also handle non-monotone risks with an extension of conformal risk control, in high probability, called Learn then Test (LTT). The main difference between this procedure and conformal risk control are twofold. First, the space of $\lambda$, denoted $\Lambda$, must be discrete. Second, the guarantee in (4) holds in high

probability over the $n$ calibration points. However, the algorithms have a roughly similar flavor; the variant of LTT we employ in all our experiments is simply a different way of setting $\hat{\lambda}$, and takes the form

$$\hat{\lambda} = \inf\{\lambda \in \Lambda : R^{+}(\lambda) \le \alpha\}, \quad (10)$$

using any concentration bound $\mathbb{P}(R^{+}(\lambda) > \mathbb{E}[\ell(X, \mathcal{C}_\lambda(X))]) \le \delta$ for all $\lambda$. We defer further detail on this procedure to[22].

**Assigning probability to hits: isotonic regression and Venn-Abers predictors.** The next method we present is geared towards a different goal: calibration. That is, given a protein $x$ and a retrieval $z$, we would like to produce a probability that the retrieval is correct (i.e. there is a functional match). We allow for any notion of correctness, parameterizing a function correct$(x, z) \in \{0, 1\}$; the indicator of an exact match is one possible choice of correct. The formal goal is to produce a probability $\hat{p}$ satisfying $\mathbb{P}(\text{correct}(x, z) = 1|\hat{p}) = \hat{p}$. This technique is useful when we have a query protein $x$, a set of known/labeled retrievals for that protein, and we seek uncertainty quantification on one or many unlabeled potential retrievals.

In this section, we develop our techniques in the setting that each protein $X$ has a set of labeled retrieval proteins, $Z^{(1)}, \ldots, Z^{(m)}$. Then, we would like to predict the probability of correct$(X, Z) = 1$ on a new, unlabeled retrieval protein $Z$. We also assume we have a pre-fit estimate of this quantity, such as the confidence of a retrieval system. This retrieval system estimate may not be any good, and its probabilities may be very uncalibrated. Our goal is to calibrate these probability estimates. We denote them as $\widehat{\text{correct}}_i = \widehat{\text{correct}}(X, Z^{(i)})$, and assume without loss of generality that they are sorted.

The canonical method for correcting these probabilities is called isotonic regression[54]. Isotonic regression solves the following optimization problem:

$$\underset{\hat{p}_1, \ldots, \hat{p}_n}{\text{minimize}} \sum_{i=1}^{n} (\hat{p}_i - \text{correct}_i)^2 \quad (11)$$
$$\text{subject to } \hat{p}_1 \le \ldots \le \hat{p}_n,$$

for any sequence $\hat{p}_1, \ldots, \hat{p}_n$ and any integer $n$. This is a convex problem admitting simple $\mathcal{O}(n)$-complexity solutions (using algorithms first introduced by[55] and[56]). Once the sequence $\hat{p}_1, \ldots, \hat{p}_n$ is found, then we can set the predicted probability of a match on the test retrieval $Z$ as

$$\hat{p} = \sum_{i=1}^{n} \hat{p}_j \mathbb{1}\left\{ \widehat{\text{correct}}(X, Z) \in \left[ \widehat{\text{correct}}_{j-1}, \widehat{\text{correct}}_j \right] \right\}, \quad (12)$$

(When $j = 0$, we set $\widehat{\text{correct}}_j = 0$.)

Under normal circumstances, the isotonic regression algorithm, given $n$ independent and identically distributed data points with $(X, Y)$, will converge to a calibrated estimate for the true, population probability of a match as $n \to \infty$. The Venn-Abers predictor[37] offers an alternative calibration strategy that does not require $n \to \infty$.

The Venn-Abers predictor works by running *two* isotonic regressions with hypothetical values correct$(X, Z)$. Namely, for $b \in \{0, 1\}$, define

$$\hat{p}_1^b, \ldots, \hat{p}_n^b, \hat{p}^b = \begin{array}{l} \underset{\hat{p}_1, \ldots, \hat{p}_n, \hat{p}}{\text{minimize}} \sum_{i=1}^{n} (\hat{p}_i - \text{correct}_i)^2 + (\hat{p} - b) \\ \text{subject to order}(\hat{p}_1, \ldots, \hat{p}_n, \hat{p}) = \text{order}\left( \widehat{\text{correct}}_1, \ldots, \widehat{\text{correct}}_n, \widehat{\text{correct}}(X, Z) \right), \end{array} \quad (13)$$

where order returns the list of indexes sorting its argument.

When the data points are exchangeable, the Venn-Abers predictor gives the following guarantee:

$$\mathbb{P}(\text{correct}(X, Z) = 1|\hat{p}^0) = \hat{p}^0 \text{ or } \mathbb{P}(\text{correct}(X, Z) = 1|\hat{p}^1) = \hat{p}^1. \quad (14)$$

Thus, the interval $[\hat{p}^0, \hat{p}^1]$ is a valid interval on the probability of correctness. One can use this strategy to report to the user that the proteins $X$ and $Z$ are, say, 70 − 73% likely to be a match.

### Dataset preparation

**SCOPe and Pfam IDs.** We select both query and lookup proteins from both UniProtKB[57] and SCOPe[45]. We use an annotated and reviewed version of UniProtKB from July 3, 2023 and select all proteins created in the database after May 25, 2022, as per[10]. The split was select to ensure there is no train-calibration leakage for the proteins used to calibrate the conformal score. This leaves 2, 350 proteins as calibration/validation, used for querying a lookup set of the remaining 540, 560 proteins. These are filtered further depending on the labels associated with the desired conformal guarantee, ie proteins annotated with Pfams. To examine guarantees across proteins with hierarchical relationships, we draw from the Astral Structural Classification of Proteins (SCOPe)[45] database version 2.08, using 40% sequence identity threshold in order to simulate the test case of remote homology. This leaves 15,177 domains in the training set across over 4693 families. For calibration, we use a test set of 400 domains, which are ensured to have < 30% sequence identity to every protein in the training set. This ensures that the model is adequately evaluated on its ability to capture features relating to remote homology. The domains in the set are filtered to have at least one other family, superfamily and fold member. For SCOPe, we calibrate on 300 proteins and test on 100 proteins

For each model and query protein $q_i$, we return an ordered list of lookup proteins $v_j$ and their similarity scores to the query as $S_{ij} = f(q_i, v_j)$, along with the associated metadata (i..e. UniProt ID) of the query and list of lookup proteins (i.e.: annotated Pfams, SCOPe IDs, organism information, etc). We generate scores $S_{ij}$ using Protein-Vec[10]; we also tested similarites derived from TOPH[11], TM-Vec[8], and Foldseek[9]. We return a rank-ordered list by similarity for each query protein $q_i$ using FAISS. With the scores $S_{ij}$ and annotations, we create pairs $\{S_{ij}, y_{ij}\}$, where $y_{ij}$ indicates a desired match in annotations. For instance, if we wish to calibrate to retrieve partial Pfam matches at a given risk (ie $q_i$ has Pfam annotation {*Pfam12345*} and $v_j$ has Pfam annotations {*Pfam12345, Pfam56789*}), we denote $y_{ij} = 1$ if there is a partial match in annotations and $y_{ij} = 0$ otherwise. This approach is particularly relevant for searching across proteins with multiple domains and/or multiple functions.

**Testing for exchangeability.** It is important to note that conformal techniques require that the data be exchangeable, else the theoretical guarantees provided are invalid. No dataset is exactly exchangeable in practice. However, we have included validations of the exchangeability assumption. In particular, we seek to show that the losses of the data are exchangeable. We test for exchangeablility of the data in the following manner:

- We split the data across timeframe. Specifically we calibrate on timeframe 2022-05-25 to 2022-12-14 and testing on 2022-12-14 to 2023-06-28. This yields 870 and 994 labeled proteins to test against the lookup set, respectively.
- We examine the cumulative density function (CDF) of FNR and FDR between samples from each timeframe and the lookup dataset.

These results are shown in Supplementary Figs. 2, 3 for FDR and FNR control, respectively. We observe the CDFs of the losses across time nearly overlap, indicating evidence of approximate exchangeability. Although it does not hold exactly, it holds to a reasonable extent, and we believe that this is sufficient to justify the use of the method.

**Enzyme classification (EC) within UniProt.** Similarly to before, we use an annotated and reviewed version of UniProtKB from July 3, 2023 and

select all proteins created in the database after May 25, 2022 as our evaluation queries $q_i$. We filter our lookup and query/evaluation sets Uniprot to only choose proteins with fully characterized EC numbers (ex: containing the full hierarchy $'a.b.c.d'$), returning a lookup set size of 211720 and a query set size of 438. This forms our distance matrix of size $(438 \times 211720)$ which we calibrate with conformal risk control using the hierarchical loss function $\text{match}(x, y) = \{0, 1, 2, 3, 4\}$ as described in (Sec 5). For the EC task, we calibrate on 300 and test on the remaining 138 proteins.

**CLEAN enzyme classification preparation.** For the generalizability test, we calibrate on $|Q| = 380$ query points of the 392 provided by *New* to maximize the calibration dataset size,. We report test precision score, recall score, F1-score, and area under curve (AUC) in addition to the hierarchical loss coverage on *Price-149*, following the metrics reported by CLEAN[39]. Thus, we assemble distograms for both *New-392* and *Price-148* against all EC cluster embeddings in CLEAN's training set, forming size $(|Q|, 5242)$ matrices where $|Q|$ represents the number of query proteins. Here, we use euclidean distances for embedding comparisons (smaller $\lambda \rightarrow$ smaller retrieval set), as done in the original work, and compute per-query hierarchical loss scores to calibrate with conformal risk control. EC numbers are normally classified in a hierarchy vector $h(x) = (A, B, C, D)$ where $(A, B, C, D)$ refer to enzyme class, subclass, sub-subclass, and serial number respectively. It is worth noting that some enzymes may be missing labels at the lower level of the hierarchy; these are denoted as * (for instance 2.3. 1. * or 2.3. *. * as Acyltransferases without further characterization). We use the hierarchical loss $\text{match}(x, y) = \{0, 1, 2, 3, 4\}$, where $\text{match}(x, y) = 0$ if $h(x) = h(y)$ exactly (two enzymes share the same class, subclass, sub-subclass, and serial number) and $\text{match}(x, y) = 4$ if $h(x)[0] \neq h(y)[0]$ (two enzymes are from different classes altogether). We note that this choice of hierarchical loss is arbitrary and can be tuned to specific application areas; for instance one may choose loss $\text{match}(x, y) = \{0, 1, 2, 4, 100\}$ to significantly penalize class-level mismatches for enzyme functional annotation.

We calibrate at $\alpha = 1$ for the *New* test benchmark, and $\alpha = 1.5$ for the *Price* generalizability benchmark. We chose a different loss threshold, $\alpha = 1$, primarily because in the hierarchical classification scheme provided by EC, the finest-resolution 4th digit is often a "serial number", differentiating between different enzymes having similar function, on the basis of the actual substrate in the reaction. In this regime, serial number errors may be somewhat tolerable to the experimentalist, whereas sub-subclass errors are less so. Thus, for a dataset with a distribution shift from *New*, such as *Price*, one may want to be more *tolerant* in the retrieval process and calibrate the model to consider mismatches 50% of the time at the sub-subclass level, and 50% of the time at the finest sub-subclass level (producing an expected $\alpha = 1.5$), to accurately obtain all enzymes of similar function. Additionally, many "preliminary" annotated enzymes exist with specificity only to the sub-subclass, so encouraging partial matches is important to attenuate the false negative rate (FNR), implicitly. We note that each query protein in *New* and *Price* may have multiple possible valid EC assignments. Thus, given a set of possible labels for each query protein $x$, we take the minimum of the hierarchical losses computed against this set of possible labels and the retrieved set of EC label centroids. Here, if a test enzyme $x$ has $k$ valid EC assignments $(x_1, \ldots, x_k)$, we want to evaluate retrieval using the assignment that produces the minimum hierarchical loss against the returned set. We do this to not unfairly penalize the model as long as it is retrieving EC cluster centroids that are close in hierarchy to one possible assignment.

**Preprocessing of diverse Dali folds across the proteome.** We extracted all 73 entries classified as "multidomain" from the Structural Classification of Proteins−extended database (SCOPe)[45]. Each entry's corresponding Protein Data Bank (PDB) file was retrieved using its

unique PDB identifier (PDBID) from the Research Collaboratory for Structural Bioinformatics (RCSB) PDB database. These files were then processed into the DALI compatible format using the built-in import function (`import.pl`) of the DALI software[2].

We utilized 73 of these chains (one was omitted as it was too short for DALI) as query structures to search against a specialized database of 2.3 million non-singleton structure representatives from the Foldseek Clustered AlphaFold database, as described in[43]. To enhance the efficiency of the DALI search process, the database was divided into multiple batches, each containing 1000 structures. The searches were conducted in parallel across these batches using multiple threads to optimize computational resource usage and decrease total runtime.

Similarly, we constructed the equivalent query and lookup set for both Protein-Vec and TM-Vec. We embed both the entire sequence database of the 2.3M Foldseek Clustered Alphafold database, as well as the 73 "multidomain" SCOPe entries, on a single A6000 GPU. Using FAISS, we index and generate similarity scores for the entire query $\times$ lookup set.

### Reporting summary
Further information on research design is available in the Nature Portfolio Reporting Summary linked to this article.

## Data availability
The associated data generated in this study to reproduce all visualizations, by figure, have been deposited in the following repository: https://zenodo.org/records/14272215. All original datasets relating to the CLEAN enzyme annotation tasks, and training data to compute embedding similarities against are available in the original source repository: https://github.com/tttianhao/CLEAN Source data are provided with this paper.

## Code availability
Code for all computational analysis, conformal prediction, algorithms, and figures is available at https://github.com/ronboger/conformal-protein-retrieval/under an Apache-2.0 license, and on Zenodo at https://doi.org/10.5281/zenodo.14279501[58].

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

## Acknowledgements

We thank members of the Doudna lab and the Innovative Genomics Institute for helpful discussions. We thank UCSF for giving us access to the high performance compute cluster Wynton to meet our compute needs. We acknowledge Dr. Daniel Bellieny Rabelo for helping run DALI on the Wynton compute cluster and Dr. Benjamin A. Adler, Dr. Jason Nomburg, Kenneth M. Loi, and Marena Trinidad for helpful feedback on the manuscript. RSB thanks the Henry Wheeler Center for Emerging and Neglected Diseases (CEND) at UC Berkeley for the Thomas C. Alber Science & Engineering for Global Health fellowship. SC thanks the Masason Foundation, the Mercatus Center for the Emergent Ventures Fellowship, and New Science for the Computational Life Sciences microgrant. ANA thanks the National Science Foundation (NSF) for the Graduate Research Fellowships Program and the Berkeley Fellowship. PHY thanks the National Science Foundation (NSF) for the Graduate Research Fellowships Program. J.A.D. is an investigator of the Howard Hughes Medical Institute, and research in the Doudna laboratory is supported by the Howard Hughes Medical Institute (HHMI), NIH/NIAID (U54AI170792, U19AI135990, UH3AI150552 and U01AI142817), NIH/NINDS (U19NS132303), NIH/NHLBI (R21HL173710), NSF (2334028), DOE (DE-AC02-05CH11231, 2553571 and B656358), Lawrence Livermore National Laboratory, Apple Tree Partners (24180), UCB-Hampton University Summer Program, Mr. Li Ka Shing, Koret-Berkeley-TAU, Emerson Collective and the Innovative Genomics Institute (IGI).

## Author contributions

R.S.B., A.N.A., S.C., and J.A.D. designed and conceived this project. R.S.B., S.C., and A.N.A. generated all introductory figures for study design and motivation. R.S.B. and A.N.A. designed, performed all computational work, and generated all figures for risk control on genes of unknown function. S.C. and R.S.B. designed, and performed all computational work and figure generation around improved enzyme function prediction and hierarchical risk control. P.H.Y. and R.S.B. designed the structural alignment experiments around the clustered AlphaFold database. R.S.B. and S.C. performed all computational analysis and figure generation. R.S.B., A.N.A., S.C., P.H.Y. wrote the methods section, detailing mathematics and data acquisition. R.S.B., A.N.A., S.C., M.I.J. contributed to the mathematical formulations in the work. All authors wrote the original draft of the manuscript, reviewed and edited the manuscript, and supported its conclusions.

## Competing interests

J.A.D. is a co-founder of Caribou Biosciences, Editas Medicine, Intellia Therapeutics, Mammoth Biosciences and Scribe Therapeutics, and a director of Altos, Johnson & Johnson and Tempus. J.A.D. is a scientific advisor to Caribou Biosciences, Intellia Therapeutics, Mammoth Biosciences, Inari, Scribe Therapeutics, Felix Biosciences and Algen. J.A.D. also serves as Chief Science Advisor to Sixth Street and a Scientific Advisory Board member at The Column Group. J.A.D. conducts academic research projects sponsored by Roche and Apple Tree Partners. The remaining authors declare no competing interests.
