## [Transparent Peer Review file · Nature Communications]

Functional protein mining with conformal guarantees

Corresponding Author: Dr Jennifer Doudna

Version 0:

Reviewer comments:

Reviewer #1

(Remarks to the Author)

This paper has the laudable goal of introducing confidence indices into the output of machine learning algorithms using the techniques of conformal prediction. As such, the work should ultimately be published, but there are a number of important issues that need to be addressed prior to publication.

1. It would really help the reader if the abstract and paper itself simply stated that the sophisticated statistical tools are designed to provide a confidence index (e.g predicted precision given a ML score or whatever other metric is used) rather than the raw ML score that the prediction is correct. This is not the first time this has been done, although this is by far a more sophisticated method, see for example H. Zhou, H. Cao and J. Skolnick. FRAGSITE: A Fragment Based Approach for Virtual Ligand Screening. Journal of Chemical Information and Modeling. 2021: 61(4): 2074-2089.

2. There is a fundamental misconception that detection of homology and or similar structure allows one to make functional inference. While this is sometimes true, it is not true in general. There are EC numbers where the EC 4th digit cannot be confidently transferred from the protein of known to unknown function even when their pairwise sequence identity is >90%.

3. Similarly, when can find pairs of structures in the PDB whose TM-score>0.5 that are very related to each other whose sequence identity<10% with entirely different function. Thus, structural similarity and functional similarity are not the same.

4. The enzyme prediction results are encouraging but essential details are missing to truly assess the significance. What is the distribution of EC numbers? How many proteins have high sequence identity between proteins of known/unknown function? On known cases, what is the False Discovery Rate? More importantly, how does the method perform in benchmarking not on a case (*Mycoplasma mycoides*) where the absolute truth is not known, but on a gold standard set with a diverse collection of known EC numbers so that one can objectively assess performance?

5. The idea of using this method to prefilter structural databases is a good one.

Overall, this is a significant contribution that should be published once the points delineated above are addressed.

(Remarks on code availability)

Reviewer #2

(Remarks to the Author)

The authors describe a general set of method to better calibrate error in several applications of ML to biological sequence analysis. They provide several great examples that include: sequence homology detection, function annotation and detecting structure-structure similarities that are found in evolutionarily distant homologs. Comment below are minor / for discussion. I am strongly positive about this paper.

The work is dependent on underlying methods, such as Prot-vec and CLEAN, but I do not think this diminishes the work, as

the authors test the method in multiple settings and demonstrate the required generality.

Perhaps add a sentence or two to the introduction that explains that distant and meaningful homologies are missed due to substantive challenges (not just computational feasibility), such as very long evo distances that induce long branch effects, HGT and other events that recombine protein part and mess up genomic context, etc. Some estimate that we cannot detect more than 1/2 of 'real' homology relationships.

It is interesting to note that most sequence homology methods only report e-values better (more surprising than) $1e^{-5}$. Thus, having all outputs between $p = 0.9995$ and 1 is common, and in many cases an output design choice (to keep proteins with many homologs from having prohibitively large outputs). Is this an output choice or an error estimation shortcoming for Prot- and TM-vec? Perhaps discuss this with a bit more nuance?

How do you deal with proteins with multiple domains and/or multiple functions? Perhaps discuss this early to guide the reader a bit more. I had to go back and forth a few times to figure this out.

Figures 2 and 3 seem crowded and have elements that might have been better as tables. Check figure 3-I, I can't make perfect sense of that arrow.

When describing matching function in methods: are exact matches the only ones you want to calibrate? $\text{match}(x,y) = 0$? I would think a few other levels of the SCOP hierarchy would be useful.

Figure 5 is needed : True or False?

Can you please increase the depth of discussion of equation (5)?

(Remarks on code availability)

Reviewer #3

(Remarks to the Author)

Please see the attached report for my detailed comments.

(Remarks on code availability)

Version 1:

Reviewer comments:

Reviewer #1

(Remarks to the Author)

The revisions have addressed all my previous concerns, and the paper is most suitable for publication in its present form.

(Remarks on code availability)

Reviewer #3

(Remarks to the Author)

I have reviewed the revised manuscript and appreciate the authors' efforts in addressing the concerns raised in the previous round. Below, I summarize my comments on the changes made, as well as suggestions for further improvement. My comments are very minor.

First, the authors have successfully clarified the overall goal of the paper, making it easier for readers to grasp the main objectives. They have also provided a well-reasoned explanation of why the multiplicity issue may not be a critical concern in this context. The revisions to the notations have reduced confusion and improved the readability of the paper.

Second, the authors have made efforts in validating the exchangeability condition through real data analysis. While this aspect is generally satisfactory, I recommend that the authors include brief explanations distinguishing risk exchangeability from the exchangeability of data points. Although risk exchangeability has been validated in the analysis, the methodology relies on the latter concept. A clearer connection between these two concepts would enhance the reader's understanding and the robustness of the methodology presented.

The other issues I raised in my previous report are relatively minor, and I believe the authors have addressed most of them effectively.

Summary: I would appreciate a brief explanation of the term *risk exchangeability*, as I find its exact meaning unclear in the context of this paper.

(Remarks on code availability)

Response to the Reviews

“Functional Protein Mining with Conformal Guarantees”

Overview

We heartily thank the three reviewers for their careful review of our manuscript. The many thoughtful comments and questions have led to a clearer presentation, for which we are very grateful.

In this response document, we begin with a summary of the structural changes to the revised manuscript. After that, we proceed to a point-by-point response to each of the comments raised. For ease of review, text quoted from the reviewers appears quoted in italics, such as this *“quote from review 1”* and our changes to the manuscript are pointed out with blue text.

Reviewer 1

Reviewer 1 summary

This paper has the laudable goal of introducing confidence indices into the output of machine learning algorithms using the techniques of conformal prediction. As such, the work should ultimately be published, but there are a number of important issues that need to be addressed prior to publication.

...

Overall, this is a significant contribution that should be published once the points delineated above are addressed.

Our response

We're extremely pleased to hear that the reviewer found our investigation interesting. We address the important issues in the revisions.

Reviewer #1, comment #1

It would really help the reader if the abstract and paper itself simply stated that the sophisticated statistical tools are designed to provide a confidence index (e.g predicted precision given a ML score or whatever other metric is used) rather than the raw ML score that the prediction is correct. This is not the first time this has been done, although this is by far a more sophisticated method, see for example H. Zhou, H. Cao and J. Skolnick. FRAGSITE: A Fragment Based Approach for Virtual Ligand Screening. Journal of Chemical Information and Modeling. 2021: 61(4): 2074-2089.

Our response #1.1

This is a good suggestion, we have made it more explicit in the abstract and the introduction that we provide a confidence index alongside raw ML scores, and we cite Zhou et al. (2021) as suggested.

Reviewer #1, comment #2

There is a fundamental misconception that detection of homology and or similar structure allows one to make functional inference. While this is sometimes true, it is not true in general. There are EC numbers where the EC 4th digit cannot be confidently transferred from the protein of known to unknown function even when their pairwise sequence identity is > 90%.

Our response #1.2

We add this point in the introduction, and we now cite Gerlt and Babbitt (2000); Whisstock and Lesk (2003).

Reviewer #1, comment #3

Similarly, when can find pairs of structures in the PDB whose TM-score>0.5 that are very related to each other whose sequence identity<10% with entirely different function. Thus, structural similarity and functional similarity are not the same.

Our response #1.3

Thank you for bringing up this point, we mention this now in the introduction.

Reviewer #1, comment #4

The enzyme prediction results are encouraging but essential details are missing to truly assess the significance. What is the distribution of EC numbers?

Our response #1.4

We added the distribution of EC numbers in supplementary figure SS4.

Reviewer #1, comment #5

How many proteins have high sequence identity between proteins of known/unknown function?

Our response #1.5

We added a histogram that shows sequence identity between Price, New test sets and the lookup dataset for the enzyme classification task in the supplemental material SS5.

Reviewer #1, comment #6

*In known cases, what is the False Discovery Rate? More importantly, how does the method perform in benchmarking not on a case (*Mycoplasma mycoides*) where the absolute truth is not known, but on a gold standard set with a diverse collection of known EC numbers so that one can objectively assess performance?*

Our response #1.6

With respect to benchmarking, we performed an analysis evaluating performance on a gold standard annotated dataset (UniProt) both for Pfams and EC numbers. This is reflected in Figs. 2B-F, 2H, Fig. SS1A for Pfam and Fig. SS7D-F for EC numbers. We also conduct rigorous benchmarks of the conformal procedure on a holdout SCOPe set of 400 domains, which are ensured to have < 30% sequence identity to every protein in the Protein-Vec training set (see Figs. 5A-C where we demonstrate hierarchical loss risk control on SCOPe). We add a histogram for the distribution of EC numbers (Fig. SS1), highlighting the data imbalance within the original CLEAN training dataset, where 4498 of the 5242 EC annotations in CLEAN have less than 50 protein examples. We believe that this highlights the merit of a calibration technique accounting for the hierarchical nature of EC assignments, where a model may not achieve the finest-level annotation, but has enough reference examples to assign an annotation within the same subfamily.

Reviewer #1, comment #7

The idea of using this method to prefilter structural databases is a good one.

Our response #1.7

Thank you!

Reviewer 2

Reviewer 2 summary

The authors describe a general set of method to better calibrate error in several applications of ML to biological sequence analysis. They provide several great examples that include: sequence homology detection, function annotation and detecting structure-structure similarities that are found in evolutionarily distant homologs. Comments below are minor / for discussion. I am strongly positive about this paper.

Our response

We are very pleased that the reviewer finds the topic of our work to be important and interesting, and we thank the reviewer for their efforts to help us improve the manuscript. The suggestions were very valuable, and we have taken care to address the points brought up by the reviewer in the revision of our manuscript.

Reviewer #2, comment #1

The work is dependent on underlying methods, such as Prot-vec and CLEAN, but I do not think this diminishes the work, as the authors test the method in multiple settings and demonstrate the required generality.

Perhaps add a sentence or two to the introduction that explains that distant and meaningful homologies are missed due to substantive challenges (not just computational feasibility), such as very long evo distances that induce long branch effects, HGT and other events that recombine protein part and mess up genomic context, etc. Some estimate that we cannot detect more than 1/2 of 'real' homology relationships

Our response #2.1

Good point, we have added this into the introduction and cite Brenner et al. (1998).

Reviewer #2, comment #2

It is interesting to note that most sequence homology methods only report e-vals better (more surprising than) $1e^{-5}$. Thus, having all outputs between $p = 0.9995$ and 1 is common, and in many cases an output design choice (to keep proteins with many homologs from having prohibitively large outputs). Is this an output choice or an error estimation shortcoming for Prot- and TM-vec? Perhaps discuss this with a bit more nuance?

Our response #2.2

We don't see this as an output choice; different embedding based protein search and annotation models will have different ranges for embedding similarity. For instance, TM-Vec, a similar method for embedding similarity search of proteins, has a range that closely resembles that of TM-Align scores.

Reviewer #2, comment #3

How do you deal with proteins with multiple domains and/or multiple functions? Perhaps discuss this early to guide the reader a bit more. I had to go back and forth a few times to figure this out.

Our response #2.3

This is a great and highly relevant question, particularly for the discovery of multidomain proteins like CRISPR-Cas systems. We take the following approach: Similarly to SS2.2.1 SCOPe and Pfam IDs Dataset Preparation (which is used for Section 3.1: Annotation of genes of unknown function with limited false discovery), we use Pfam annotated proteins in UniProt. Pairs of proteins with a partial Pfam match (for instance x_1 : *Pfam12345*, x_2 : *Pfam12345*, *Pfam56789*) in addition to exact Pfam matches (for instance x_1 : *Pfam12345*, x_2 : *Pfam12345*) are given a positive label; proteins without a match are given a negative label. Using the similarity score between pairs of proteins and the match label (we denote this pair s_i, y_i), we calibrate across multiple trials with both conformal risk control and learn then test methods depending on the desired loss (ie FDR, FNR) to determine a threshold value λ . Determining this threshold, we can then do discovery of partial and exact matches across proteins for multiple domains. We clarify these details in the methods section.

With the scores S_{ij} and annotations, we create pairs $\{S_{ij}, y_{ij}\}$, where y_{ij} indicates a desired match in annotations. For instance, if we wish to calibrate to retrieve partial Pfam matches at a given risk (ie q_i has Pfam annotation $\{Pfam12345\}$ and v_j has Pfam annotations $\{Pfam12345, Pfam56789\}$), we denote $y_{ij} = 1$ if there is a partial match in annotations and $y_{ij} = 0$ otherwise. This approach is particularly relevant for searching across proteins with multiple domains and/or multiple functions.

Reviewer #2, comment #4

Figures 2 and 3 seem crowded and have elements that might have been better as tables. Check figure 3-I, I can't make perfect sense of that arrow.

Our response #2.4

Agreed, there is a lot of information to convey about the method! This is good feedback to improve the clarify of the manuscript; we made the following changes:

- For Fig. 2, we move the 2I and 2J to the supplement.
- For Fig. 3, we move 3B, 3C, 3D into the supplement. Figure 3-A, 3-B, 3-D, 3-E is all information that is hypothetically captured in Table1; however, moving all reduces emphasis on performance.
- The arrow in Fig. 3I reflects the point of max-separation, as defined by Yu et al 2023, Science. We make this more clear in the revised manuscript's Fig. 3 caption.

Reviewer #2, comment #5

When describing matching function in methods: are exact matches the only ones you want to calibrate? $\text{match}(x, y) = 0$? I would think a few other levels of the SCOP hierarchy would be useful.

Our response #2.5

Good point, we address this by increasing the $\text{match}(x, y)$ score. For instance, $\text{match}(x, y) = 1$ would allow for superfamily matches but family level mismatches. If you are referring to multidomain proteins, we address how to calibrate search in a previous comment. We clarify both in the manuscript.

Reviewer #2, comment #6

Figure 5 is needed : True or False?

Our response #2.6

We moved this figure and subsection into the supplement for those looking for more examples for hierarchical loss with common benchmarks such as SCOPe and EC numbers on UniProt.

Reviewer #2, comment #7

Can you please increase the depth of discussion of equation (5)?

Our response #2.7

We included the following paragraph under equation (5) in section S2.1.1 (Conformal risk control), which we hope will make things intuitively clear:

The calibration procedure for $\hat{\lambda}$ is doing something simple and easy to understand. On the left-hand side of the inequality in (Eq. (4)), we have the empirical risk, i.e., the average loss on our calibration data. On the right-hand side of the inequality, we have α minus a small fudge-factor that decays as $1/n$. Thus, we are picking the smallest λ —often indicating the smallest retrieval set—such that the risk is bounded above by α (fudge-factor aside). Importantly, the loss in (4) is not any specific loss, like the false negative rate or false discovery rate—it is a general, bounded loss (although non-monotone losses have a slightly different algorithm; see Angelopoulos et al. (2021))

Reviewer 3

Main Response to Reviewer #3

We are very grateful to have had a clearly expert reviewer review our manuscript so thoroughly. The comments have been extraordinarily helpful in helping us improve our manuscript.

The primary concern of this reviewer is in the mathematical notation used. The notation had some initial inconsistencies and typos, which we have fixed. We apologize and are grateful to the reviewer for catching these.

These notational issues seem to have caused some statistical misunderstandings related to a) the problem setup, and b) multiplicity, which we seek to clarify before getting into point-by-point responses.

Problem Setup

We expect biology practitioners to use our method for the following prediction problem:

1. The practitioner has a protein of interest, such as GFP. They would like to use machine learning to find other proteins with similar function. We call this protein the *query protein*.
2. The practitioner uses our algorithm to search that protein up against a lookup database D .
3. Our algorithm returns $C_{\hat{\lambda}}$, a subset of D .

Hopefully this clarifies the goal of our paper: for each query protein, separately, we would like to retrieve a set of similar proteins (with statistical guarantees).

The role of Q is only to provide a validation set of query proteins, for the purpose of testing our methodology. This is similar to how we might measure any learning method's per-data-point performance on a large validation set.

The confusion may have been that we did not index C_{λ} by $q \in Q$, which made it look like we used the whole Q in order to produce the set. To address this, we now use the notation $C_{\lambda}(q)$ to denote the retrieval set produced for query protein q .

Multiplicity

It is possible that the reviewer's concern about multiplicity will be entirely alleviated by the above description of the problem setup.

In particular, there is no further multiplicity to account for in this procedure. For each $q \in Q$, we produce a set, $C_{\hat{\lambda}}(q)$. This set comes with a guarantee that $E[\ell(q, C_{\hat{\lambda}}(q))] \leq \alpha$. This is a *single guarantee on the set* $C_{\hat{\lambda}}(q)$, per q ; i.e., there is only *one* test being run, not multiple. The guarantee is not simultaneous over all $q \in Q$, nor does it incur any multiplicity in D ; it is a single guarantee on the set, $C_{\hat{\lambda}}(q)$. For more detail on this family of approaches, see Angelopoulos et al. (2022) (we suspect the reviewer may already be familiar with these approaches, but we include the reference just in case).

Further Response

Reviewer 3 summary

The article presents a class of conformal methods aimed at comparing multiple query proteins against an extensive lookup library of proteins for further characterization. This research addresses a significant and challenging problem in the field, motivated by the expansive size of

protein datasets, the insufficiency of traditional statistical methods, the inherent complexity of protein structures, and the reliance on approaches based on black-box models. The core issues tackled in this research include the dual aspects of discovery and pre-filtering in the selection process. Specifically, the authors address critical issues such as the control of the false discovery rate (FDR) and the false negative rate (FNR). The proposed conformal approaches offer a promising and logical framework for overcoming the aforementioned challenges.

Strengths: The strengths of the work include (a) an innovative approach with a solid statistical foundation and (b) the relevance of the proposal, featuring a well-integrated set of methods and applications. Specifically, the integration of conformal methods in protein comparison is a valuable contribution, providing statistical guarantees that enhance the reliability and uncertainty quantification in the protein characterization process. Moreover, the flexible nature of the proposed methods is particularly relevant in light of the current challenges faced in bioinformatics, highlighting the need for advanced statistical tools.

Weaknesses: The current paper has several significant drawbacks that clearly require improvements. Based on my field expertise, I will focus on the related statistical issues and their connection to application. The points listed below will be expanded and further explained in the "Comments" section.

First, the methodological clarity needs to be improved. Several key issues are not adequately explained, which may hinder reproducibility and understanding. A more detailed description of the conformal methods and their implementation is necessary.

Second, the notation lacks consistency. The manuscript exhibits inconsistencies in notation that may confuse readers. A systematic enhancement of definitions and terms would improve clarity.

Third, there is insufficient explanation to bridge the gap between statistical methods and application; more details should be provided to enhance context and understanding for the reader.

Our response

The above 3 weaknesses are addressed in the Main Response above as well as further text below.

Finally, there is a lack of validation for key assumptions. The manuscript noticeably lacks validation for the key assumptions made in the analysis. The authors should provide empirical evidence or theoretical justification to support these assumptions, thereby strengthening the overall argument.

Our response

The point of exchangeability is a good one; no dataset is exactly exchangeable in practice. However, we have included validations of the exchangeability assumption. Although it does not hold exactly, it holds to a reasonable extent, and we believe that this is sufficient to justify the use of the method. In particular, the risks are approximately exchangeable, based on our analysis, even after splitting *in time* (i.e., calibrating on timeframe 2022-05-25 to 2022-12-14 and testing on 2022-12-14 to 2023-06-28). See the new Figure S2-3 for details.

Conclusion: The manuscript requires substantial revisions for clarity, consistency, and validation of its methodologies. The overall writing quality is not very high. Addressing these issues is essential before the manuscript can be considered for publication.

Our response

We thank the reviewer for their feedback. We revised the manuscript for clarity, consistency, and validation of methodologies accordingly. We also stressed improvement on the writing quality of the manuscript.

Reviewer #3, comment #1

P1. The claim that one can “select any loss metric” is overly broad. I do not think that conformal methods can handle any loss metric. This kind of statement does not seem to be professional, and should be toned down and moderated for clarity.

Our response #3.1

The family of conformal methods that we use, including Conformal Risk Control (Angelopoulos et al. (2022)) and Learn then Test (Angelopoulos et al. (2021)), can handle any bounded loss in finite samples.

We changed this text to “many loss metrics of biological relevance,” because we do not expect the primary audience of this paper to know what a bounded loss is.

In the methods, we have elaborated on the exact technical conditions under which these techniques work: conformal risk control for bounded loss functions, and learn then test for general bounded loss functions (and asymptotically, any loss with a second moment).

Reviewer #3, comment #2

P3. The statement “traditional statistical methods such as e-values” is somewhat misleading, as e-values are a relatively recent concept that is closely related to conformal prediction methods?

Our response #3.2

We replaced this sentence with “These techniques address the emerging challenges that have arisen with the large scale of protein data and complex deep learning architectures, which cannot be addressed by traditional parametric statistical methods.”

Reviewer #3, comment #3

P3. EC annotation: what does EC represent?

Our response #3.3

EC represents an “Enzyme Commission number,” a numerical classification scheme for enzymes, based on the chemical reactions they catalyze - we define this in the first paragraph section 3.2. We change EC to enzyme function annotation in P3 accordingly to avoid confusion during the introduction.

Reviewer #3, comment #4

P3. I find the definitions of \hat{C} and C_λ to be somewhat confusing. In their current form, both notations utilize C , yet their meanings appear to be inconsistent.

Our response #3.4

Thank you for bringing up these inconsistencies in the notation. We simplified the notation to make sure it was consistent and easier to read.

P3 The notation $\hat{C} \in D$ seems to be incorrect. Should it not be $\hat{C} \subset D$? Given that there are several renowned researchers on the author list, am I possibly overlooking or misunderstanding something? If this is merely an elementary typo, it raises concerns about the quality of this work. ...

Our response #3.4

Thank you, we changed $C \in D$ to $C \subseteq D$.

The notation $C_\lambda = \{v : S_{ij} > \lambda\}$ is also unclear. I do not understand what v represents (there is no explanation or definition of this quantity). ...

Our response #3.4

We changed the notation to be more consistent: $C_\lambda(Q_i) = D_j \in D : S_{ij} > \lambda$. Thank you.

While we have both Q and D , the definition of \hat{C} appears to focus solely on a single protein in Q . Is this appropriate? ...

Our response #3.4

Yes. The purpose of Q is only for our experiments; in practice, we expect users of this method only to apply it to a single protein. See the “Main response to Reviewer #3” for more detail here.

It seems that \hat{C} is a set of indices in $[N_D]$, while C_λ , as a potential candidate for \hat{C} , is a set of index pairs (i, j) where $i \in [N_Q]$ and $j \in [N_D]$. Is this interpretation correct? If so, the current notation system is obviously problematic. ...

Our response #3.4

This interpretation is not correct; see the “Main response to Reviewer #3”. It is possible that the initial notation caused this confusion, and we hope we have fixed the problem.

In general, multiplicities may arise from two factors: multiple proteins in the query set and multiple proteins in the lookup library. The definitions provided above do not accurately reflect the underlying structural complexities of the problem. A more careful discussion on related issues should be offered. ...

Our response #3.4

To be clear: no statistical multiplicity adjustment is needed for our questions of interest. Our target of inference is not on a per-protein basis, but an evaluation of the set as a whole. Please see “Main response to Reviewer #3”.

Reviewer #3, comment #5

P3. The sentence “Depending on the model’s ability to capture ... compared to” is awkwardly written and does not appear to be accurate.

Our response #3.5

We reworded this to:

In some cases, the model may assign lower similarity scores S_{ij} to proteins with functional similarity, while more distant proteins may receive higher scores due to model limitations.

Reviewer #3, comment #6

P4. Regarding Figure 1(A), I seek clarification on two points: (a) the detailed construction process of the calibration dataset and (b) the determination of $\hat{\lambda}$. For (a), the authors should provide a pointer where this issue is elaborated, as the current description does not effectively connect the method to its application. For (b), it is unclear which error metric has been considered. Could you be more specific about: (a) what is the criterion for distinguishing a correct discovery from a false one? (b) What error rate is employed in this context? ...

Our response #3.6

On (a): Thank you for this comment. In response, we have elaborated on the dataset construction in the methods section and provided a pointer in the figure.

On (b): Figure 1(A) is a general pipeline figure that shows the process of constructing the retrieval set. This pipeline is agnostic to the loss function: in the paper, we use both FDR and FNR, but the overall procedure is roughly the same, involving a calibration step. We hoped to define a more general pipeline that is agnostic to the particular loss, and defer these details to the methods section, since most of our readers will not be statistically-minded.

A correct discovery in our case is simply a correct retrieval (by a domain-specific definition). For example, in our experiments relating to enzyme annotation, we sometimes define a correct retrieval as an EC-Level 4 (serial number) match.

More importantly, a critical issue is not addressed anywhere in the manuscript: how is multiplicity evaluated (i.e., what is the total number of tests in the multiple testing problem): over the ND proteins in the library, over the NQ proteins in the query, or over the $ND \times NQ$ protein pairs?

Our response #3.6

See the “Main response to reviewer #3” for discussion of evaluating multiplicity. Multiplicity correction is not needed in our setting. We are only producing a single test, and there is a *single guarantee on the set* $C_{\hat{\lambda}}(q)$, per q , that $E[\ell(q, C_{\hat{\lambda}}(q))] \leq \alpha$.

Reviewer #3, comment #7

P4. What does the part crossed out with a horizontal line in Figure 1(D) mean? Is it a typo or there is a specific meaning?

Our response #3.7

The part crossed out with a horizontal line refers to an analogy describing incorrect annotation assignments. We now clarify this further in the revised manuscript Figure 1(D), where we show an example of an retrieved annotation with hierarchical loss 1 (indicating a finest-level family mismatch), between methionine S-methyltransferase (EC 2.1.1.12) and methionine synthase (EC 2.1.1.13). This should make the hierarchical tree to the right in 1(D) more intuitive as to the loss we employ.

Reviewer #3, comment #8

There is evidence that many sentences appear to be directly taken from ChatGPT without any corrections/refinements. For example, the whole phrase “false negative rate (FNR)” is used multiple times throughout the text. This must be changed automatically by the machine. Any human being would only stated this once, and use “FNR” thereafter. I found this quite annoying.

Our response #3.8

We used ChatGPT for occasional revisions in writing the manuscript, but wrote the text ourselves. We earnestly restated “false negative rate (FNR)” to clarify for a more biological audience who is less familiar with statistical terminology. We removed this in the current revision.

Reviewer #3, comment #9

P5. Regarding the title of Section 3.1, I do not understand the meaning of “limited false discovery.” In general it does not appear to be a proper term from the perspective of a well-trained statistician. You should consider using ‘controlling the false discovery’ or ‘risk control,’ as these terms sound more professional and accurate.

Our response #3.9

Thank you for raising this point, to ensure accuracy of our claims we change this to “control the false discovery rate”.

Reviewer #3, comment #10

P5. When the concepts of false discovery rate (FDR) and false negative rate (FNR) are first introduced, it is important to provide appropriate references. Notably, there is no consensus regarding the definition of FNR. In the literature, similar terms such as false negative rate, false non-discovery rate, and missed discovery rate have been used but their meanings often differ in different papers. Therefore, a precise definition should be included.

Our response #3.10

This is an excellent point, we add our definitions for FNR and FDR in the revised manuscript in the supplementary material; see the below paragraph:

“For clarity, we define some of the commonly utilized loss functions for our retrieved sets, false discovery rate (FDR) and false negative rate (FNR). Motivated by the desire to control against false significant hits, we define false discovery rate first. FDR measures the ratio between false positive protein hits (false discoveries) in our retrieved set of model-derived significant hits to the total number of hits (the size of our retrieved set. The FNR, similarly, is the number of false significant hits as a fraction of the total pool of possible hits.”

Reviewer #3, comment #11

P5. The notations \hat{p}^0 and \hat{p}^1 are not defined, and the related discussions are somewhat confusing.

Our response #3.11

We significantly simplified the discussion of the Venn-Abers predictor in the main text, and referred the reader to the supplementary methods section. (It is difficult to simply explain the Venn-Abers predictor to a broad audience.)

Reviewer #3, comment #12

The notations in the paper are inconsistent and often introduced without proper explanations. For example, on page 14, α is first defined as the FDR level and is specified as 0.1. Then, without explanation, different levels are introduced: fold-level $\alpha = 1$ and superfamily-level $\alpha = 2$. I am unclear about the meaning of these α values and how they can be greater than 1?

Our response #3.12

We performed experiments on two well-known structural bioinformatics benchmarks in this section. In both of these experiments, we use a hierarchical setting, with a loss function taking values in $\{0, 1, 2, 3, 4\}$

depending on the mismatch in the hierarchy. This explains how $\alpha = 1$ and $\alpha = 2$ are reasonable values. We remove the $\alpha = 0.1$ to avoid confusion, as we focus on demonstrating coverage of the hierarchical loss on these two benchmarks (Enzyme Commission numbers, SCOPe).

Reviewer #3, comment #13

P15. The exchangeability assumption is fundamental when applying conformal methods, as it ensures that the calibration set and test set are representative of the same underlying distribution. However, based on the data preparation and experimental design presented, there appears to be insufficient validation demonstrating that the test data are exchangeable with the calibration data.

In conformal inference, the validity of inference relies heavily on this assumption. If the calibration and test datasets are not exchangeable, the resulting conformity scores may be misleading, leading to unreliable inference. To ensure the correctness of the results, it is essential to conduct thorough evaluations using well-designed experiments in real data analysis. This could involve assessing the exchangeability through various statistical tests or employing resampling methods to compare calibration and test sets. Without such validation, the reliability of the results may be compromised, raising concerns about their trustworthiness.

Our response #3.13

Good point; no dataset is exactly exchangeable in practice. However, we have included validations of the exchangeability assumption. Although it does not hold exactly, it holds to a reasonable extent, and we believe that this is sufficient to justify the use of the method. In particular, the risks are approximately exchangeable, based on our analysis, even after splitting *in time* (i.e., calibrating on timeframe 2022-05-25 to 2022-12-14 and testing on 2022-12-14 to 2023-06-28). See the new Figure S2-3 for details.

Based on our understanding of the biology, it is not surprising that we observe approximate exchangeability. Protein sequences and structures tend to evolve slowly and share conserved elements across species, which maintains stability in homology relationships even when samples are taken across different time frames. Biological sampling (and as a result protein language models, see Ding and Steinhardt (2024)) tend to be biased towards organisms of interest to humans. Given the structured nature of protein evolution and capturing similar organisms over time, it is not inconceivable that the loss between samples across time be roughly exchangeable.

Reviewer #3, comment #14

P15. In equation (5), it is unclear whether the FDR or the FNR is utilized. The methods should differ, which would lead to different values of $\hat{\lambda}$.

Our response #3.14

This equation is not for a specific loss. It is for a general loss function.

Another reviewer asked for an intuitive explanation of this equation, so we included the following paragraph:

The calibration procedure for $\hat{\lambda}$ is doing something simple and easy to understand. On the left-hand side of the inequality in (5), we have the empirical risk, i.e., the average loss on our calibration data. On the right-hand side of the inequality, we have α minus a small fudge-factor that decays as $1/n$. Thus, we are picking the smallest λ —often indicating the smallest retrieval set—such that the risk is bounded above by α (fudge-factor aside). Importantly, the loss in (5) is not any specific loss, like the false negative rate or false discovery rate—it is a general, bounded loss (although non-monotone losses have a slightly different algorithm; see Angelopoulos et al. (2021)).

Reviewer #3, comment #15

P15. The sections are numbered in a rather unconventional manner, which gives rise to an inconsistency in the document's structure. Commencing with a subsection without a preceding subsection appears unconventional and can potentially confuse readers. In particular, the numbering 4.0.1 stands out as unusual, as it deviates from typical hierarchical numbering conventions (at least personally, I find this type

of numbering unconventional). Considering the collaborative effort involved in authoring the paper (there are many authors), it is somewhat surprising that this oversight regarding the section numbering was not noticed or addressed by any of the contributing authors.

Our response #3.15

We revised this to have a more appropriate numbering scheme, thank you.

Reviewer #3, comment #16

P16. The optimization problem presented in (11) requires further clarification. It involves two distinct dimensions: NQ for query proteins and ND for library or dictionary proteins. Is the monotonicity adjustment applied to the m scores associated with the same query protein? Additionally, what is the relationship between m and n ? Has cross-protein adjustment been taken into account? I would appreciate it if you could provide more details about the isotonic regression problem.

Our response #3.16

Yes, the monotonicity adjustment is applied to the m scores associated to the same query protein. The n is just an arbitrary integer used to define the optimization problem; we added some sentences to clarify this, including defining the optimization problem

for any sequence $\hat{p}_1, \dots, \hat{p}_n$ and any integer n .

We also added

This technique is useful when we have a query protein x , a set of known/labeled retrievals for that protein, and we seek uncertainty quantification on one or many unlabeled potential retrievals.

Hopefully this clarifies the reviewer's confusion about cross-protein adjustment etc. No such adjustments are needed, because we are handling the case where there are some labeled retrievals for the query protein, and we simply are asking for uncertainty quantification on other unlabeled retrievals for the same query protein.

References

- Angelopoulos, A. N., Bates, S., Candès, E. J., Jordan, M. I., and Lei, L. (2021). Learn then test: Calibrating predictive algorithms to achieve risk control. *arXiv preprint arXiv:2110.01052*.
- Angelopoulos, A. N., Bates, S., Fisch, A., Lei, L., and Schuster, T. (2022). Conformal risk control. *arXiv preprint arXiv:2208.02814*.
- Brenner, S. E., Chothia, C., and Hubbard, T. J. (1998). Assessing sequence comparison methods with reliable structurally identified distant evolutionary relationships. *Proceedings of the National Academy of Sciences*, 95(11):6073–6078.
- Ding, F. and Steinhardt, J. N. (2024). Protein language models are biased by unequal sequence sampling across the tree of life. *bioRxiv*, pages 2024–03.
- Gerlt, J. A. and Babbitt, P. C. (2000). Can sequence determine function? *Genome biology*, 1:1–10.
- Whisstock, J. C. and Lesk, A. M. (2003). Prediction of protein function from protein sequence and structure. *Quarterly reviews of biophysics*, 36(3):307–340.
- Zhou, H., Cao, H., and Skolnick, J. (2021). Fragsite: a fragment-based approach for virtual ligand screening. *Journal of chemical information and modeling*, 61(4):2074–2089.

We thank the reviewers for their comments in the second round of review, and are elated to learn that they see value in the work and that the work has been accepted.

REVIEWERS' COMMENTS

Reviewer #1 (Remarks to the Author):

The revisions have addressed all my previous concerns, and the paper is most suitable for publication in its present form.

Thank you! We thank Reviewer #1 for their previous comments.

Reviewer #3 (Remarks to the Author):

I have reviewed the revised manuscript and appreciate the authors' efforts in addressing the concerns raised in the previous round. Below, I summarize my comments on the changes made, as well as suggestions for further improvement. My comments are very minor.

First, the authors have successfully clarified the overall goal of the paper, making it easier for readers to grasp the main objectives. They have also provided a well-reasoned explanation of why the multiplicity issue may not be a critical concern in this context. The revisions to the notations have reduced confusion and improved the readability of the paper.

Second, the authors have made efforts in validating the exchangeability condition through real data analysis. While this aspect is generally satisfactory, I recommend that the authors include brief explanations distinguishing risk exchangeability from the exchangeability of data points. Although risk exchangeability has been validated in the analysis, the methodology relies on the latter concept. A clearer connection between these two concepts would enhance the reader's understanding and the robustness of the methodology presented.

The other issues I raised in my previous report are relatively minor, and I believe the authors have addressed most of them effectively.

*Summary: I would appreciate a brief explanation of the term *risk exchangeability*, as I find its exact meaning unclear in the context of this paper.*

Thank you! We thank Reviewer #3 for their previous comments and suggestions, provoking a deeper discussion on the notation and mathematics of the paper.

We now expand more on "risk exchangeability" and how it is different from data exchangeability in the methods. The added text has been provided below.

The particular technical condition required for our theory is that the vector of losses $(\|X_1\|, \|C_{\lambda}(X_1)\|, \dots, \|X_n\|, \|C_{\lambda}(X_n)\|)$, $\|X$,

$C_{\lambda}(X)$ is exchangeable for all λ ---the exchangeability of the data points implies this fact, but it is technically weaker.
In other words, exchangeability matters only insofar as the risk is concerned.

Referee Report on “Functional protein mining with conformal guarantees”

1 Summary

The article presents a class of conformal methods aimed at comparing multiple query proteins against an extensive lookup library of proteins for further characterization. This research addresses a significant and challenging problem in the field, motivated by the expansive size of protein datasets, the insufficiency of traditional statistical methods, the inherent complexity of protein structures, and the reliance on approaches based on black-box models. The core issues tackled in this research include the dual aspects of discovery and pre-filtering in the selection process. Specifically, the authors address critical issues such as the control of the false discovery rate (FDR) and the false negative rate (FNR). The proposed conformal approaches offer a promising and logical framework for overcoming the aforementioned challenges.

Strengths: The strengths of the work include (a) an innovative approach with a solid statistical foundation and (b) the relevance of the proposal, featuring a well-integrated set of methods and applications. Specifically, the integration of conformal methods in protein comparison is a valuable contribution, providing statistical guarantees that enhance the reliability and uncertainty quantification in the protein characterization process. Moreover, the flexible nature of the proposed methods is particularly relevant in light of the current challenges faced in bioinformatics, highlighting the need for advanced statistical tools.

Weaknesses: The current paper has several significant drawbacks that clearly require improvements. Based on my field expertise, I will focus on the related statistical issues and their connection to application. The points listed below will be expanded and further explained in the “Comments” section.

First, the methodological clarity needs to be improved. Several key issues are not adequately explained, which may hinder reproducibility and understanding. A more detailed description of the conformal methods and their implementation is necessary.

Second, the notation lacks consistency. The manuscript exhibits inconsistencies in notation that may confuse readers. A systematic enhancement of definitions and terms would improve clarity.

Third, there is insufficient explanation to bridge the gap between statistical methods and application; more details should be provided to enhance context and understanding for the reader.

Finally, there is a lack of validation for key assumptions. The manuscript noticeably lacks validation for the key assumptions made in the analysis. The authors should provide empirical evidence or theoretical justification to support these assumptions, thereby

strengthening the overall argument.

Conclusion: The manuscript requires substantial revisions for clarity, consistency, and validation of its methodologies. The overall writing quality is not very high. Addressing these issues is essential before the manuscript can be considered for publication.

2 Specific Comments

Here is a list of my comments and concerns, organized according to the page order where the issues arise.

1. P1. The claim that one can “select any loss metric” is overly broad. I do not think that conformal methods can handle *any* loss metric. This kind of statement does not seem to be professional, and should be toned down and moderated for clarity.
2. P3. The statement “traditional statistical methods such as e-values” is somewhat misleading, as e-values are a relatively recent concept that is closely related to conformal prediction methods?
3. P3. **EC** annotation: what does EC represent?
4. P3. I find the definitions of \hat{C} and C_λ to be somewhat confusing. In their current form, both notations utilize C , yet their meanings appear to be inconsistent.
 - The notation $\hat{C} \in D$ seems to be incorrect. Should it not be $\hat{C} \subset D$? Given that there are several renowned researchers on the author list, am I possibly overlooking or misunderstanding something? If this is merely an elementary typo, it raises concerns about the quality of this work.
 - The notation $C_\lambda = \{v : S_{ij} > \lambda\}$ is also unclear. I do not understand what v represents (there is no explanation or definition of this quantity).
 - While we have both Q and D , the definition of \hat{C} appears to focus solely on a single protein in Q . Is this appropriate?
 - It seems that \hat{C} is a set of indices in $[N_D]$, while C_λ , as a potential candidate for \hat{C} , is a set of index pairs (i, j) where $i \in [N_Q]$ and $j \in [N_D]$. Is this interpretation correct? If so, the current notion system is obviously problematic.
 - In general, multiplicities may arise from two factors: multiple proteins in the query set and multiple proteins in the lookup library. The definitions provided above do not accurately reflect the underlying structural complexities of the problem. A more careful discussion on related issues should be offered.
5. P3. The sentence “Depending on the model’s ability to capture ... compared to” is awkwardly written and does not appear to be accurate.

6. P4. Regarding Figure 1(A), I seek clarification on two points: (a) the detailed construction process of the calibration dataset and (b) the determination of $\hat{\lambda}$. For (a), the authors should provide a pointer where this issue is elaborated, as the current description does not effectively connect the method to its application. For (b), it is unclear which error metric has been considered. Could you be more specific about: (a) what is the criterion for distinguishing a correct discovery from a false one? (b) What error rate is employed in this context?

More importantly, a critical issue is not addressed anywhere in the manuscript: how is multiplicity evaluated (i.e., what is the total number of tests in the multiple testing problem): over the N_D proteins in the library, over the N_Q proteins in the query, or over the $N_D \times N_Q$ protein pairs?

7. P4. What does the part crossed out with a horizontal line in Figure 1(D) mean? Is it a typo or there is a specific meaning?
8. There is evidence that many sentences appear to be directly taken from ChatGPT without any corrections/refinements. For example, the whole phrase “false negative rate (FNR)” is used multiple times throughout the text. This must be changed automatically by the machine. Any human being would only stated this once, and use “FNR” thereafter. I found this quite annoying.
9. P5. Regarding the title of Section 3.1, I do not understand the meaning of “limited false discovery.” In general it does not appear to be a proper term from the perspective of a well-trained statistician. You should consider using ‘controlling the false discovery’ or ‘risk control,’ as these terms sound more professional and accurate.
10. P5. When the concepts of false discovery rate (FDR) and false negative rate (FNR) are first introduced, it is important to provide appropriate references. Notably, there is no consensus regarding the definition of FNR. In the literature, similar terms such as false negative rate, false non-discovery rate, and missed discovery rate have been used but their meanings often differ in different papers. Therefore, a precise definition should be included.
11. P5. The notations \hat{p}^0 and \hat{p}^1 are not defined, and the related discussions are somewhat confusing.
12. The notations in the paper are inconsistent and often introduced without proper explanations. For example, on page 14, α is first defined as the FDR level and is specified as 0.1. Then, without explanation, different levels are introduced: fold-level $\alpha = 1$ and superfamily-level $\alpha = 2$. I am unclear about the meaning of these α values and how they can be greater than 1?
13. P15. The exchangeability assumption is fundamental when applying conformal methods, as it ensures that the calibration set and test set are representative of the same underlying distribution. However, based on the data preparation and experimental

design presented, there appears to be insufficient validation demonstrating that the test data are exchangeable with the calibration data.

In conformal inference, the validity of inference relies heavily on this assumption. If the calibration and test datasets are not exchangeable, the resulting conformity scores may be misleading, leading to unreliable inference. To ensure the correctness of the results, it is essential to conduct thorough evaluations using well-designed experiments in real data analysis. This could involve assessing the exchangeability through various statistical tests or employing resampling methods to compare calibration and test sets. Without such validation, the reliability of the results may be compromised, raising concerns about their trustworthiness.

14. P15. In equation (5), it is unclear whether the FDR or the FNR is utilized. The methods should differ, which would lead to different values of $\hat{\lambda}$.
15. P15. The sections are numbered in a rather unconventional manner, which gives rise to an inconsistency in the document's structure. Commencing with a subsubsection without a preceding subsection appears unconventional and can potentially confuse readers. In particular, the numbering **4.0.1** stands out as unusual, as it deviates from typical hierarchical numbering conventions (at least personally, I find this type of numbering unconventional). Considering the collaborative effort involved in authoring the paper (there are many authors), it is somewhat surprising that this oversight regarding the section numbering was not noticed or addressed by any of the contributing authors.
16. P16. The optimization problem presented in (11) requires further clarification. It involves two distinct dimensions: N_Q for query proteins and N_D for library or dictionary proteins. Is the monotonicity adjustment applied to the m scores associated with the same query protein? Additionally, what is the relationship between m and n ? Has cross-protein adjustment been taken into account? I would appreciate it if you could provide more details about the isotonic regression problem.